# Learning Kernel Tests Without Data Splitting

**Jonas M. Kübler**    **Wittawat Jitkrittum**[*]    **Bernhard Schölkopf**    **Krikamol Muandet**
Max Planck Institute for Intelligent Systems, Tübingen, Germany
{jmkuebler, bs, krikamol}@tue.mpg.de, wittawatj@gmail.com

## Abstract

Modern large-scale kernel-based tests such as maximum mean discrepancy (MMD) and kernelized Stein discrepancy (KSD) optimize kernel hyperparameters on a held-out sample via data splitting to obtain the most powerful test statistics. While data splitting results in a tractable null distribution, it suffers from a reduction in test power due to smaller test sample size. Inspired by the selective inference framework, we propose an approach that enables learning the hyperparameters and testing on the full sample without data splitting. Our approach can correctly calibrate the test in the presence of such dependency, and yield a test threshold in closed form. At the same significance level, our approach's test power is empirically larger than that of the data-splitting approach, regardless of its split proportion.

## 1   Introduction

Statistical hypothesis testing is a ubiquitous problem in numerous fields ranging from astronomy and high-energy physics to medicine and psychology [1]. Given a hypothesis about a natural phenomenon, it prescribes a systematic way to test the hypothesis empirically [2]. Two-sample testing, for instance, addresses whether two samples originate from the same process, which is instrumental in experimental science such as psychology, medicine, and economics. This procedure of rejecting false hypotheses while retaining the correct ones governs most advances in science.

Traditionally, test statistics are usually fixed prior to the testing phase. In modern-day hypothesis testing, however, practitioners often face a large family of test statistics from which the best one must be selected before performing the test. For instance, the popular kernel-based two-sample tests [3, 4] and goodness-of-fit tests [5, 6] require the specification of a kernel function and its parameter values. Abundant evidence suggests that finding good parameter values for these tests improves their performance in the testing phase [4, 7–9]. As a result, several approaches have recently been proposed to learn optimal tests directly from data using different techniques such as optimized kernels [4, 9–13], classifier two-sample tests [14, 15], and deep neural networks [16, 17], to name a few. In other words, the modern-day hypothesis testing has become a two-stage "learn-then-test" problem.

Special care must be taken in the subsequent testing when optimal tests are learned from data. If the same data is used for both learning and testing, it becomes harder to derive the asymptotic null distribution because the selected test and the data are now dependent. In this case, conducting the tests as if the test statistics are independent from the data leads to an uncontrollable false positive rate, see, e.g., our experimental results. While permutation testing can be applied [18], it is too computationally prohibitive for real-world applications. Up to now, the most prevalent solution is *data splitting*: the data is randomly split into two parts, of which the former is used for learning the test while the latter is used for testing. Although data splitting is simple and in principle leads to the correct false positive rate, its downside is a potential loss of power.

In this paper, we investigate the two-stage "learn-then-test" problem in the context of modern kernel-based tests [3–6] where the choice of kernel function and its parameters play an important role. The

---

[*]Now with Google Research

key question is *whether it is possible to employ the full sample for both learning and testing phase without data splitting, while correctly calibrating the test in the presence of such dependency*. We provide an affirmative answer if we learn the test from a vector of jointly normal base test statistics, e.g., the linear-time MMD estimates of multiple kernels. The empirical results suggest that, at the same significance level, the test power of our approach is larger than that of the data-splitting approach, regardless of the split proportion (cf. Section 5). The code for the experiments is available at https://github.com/MPI-IS/tests-wo-splitting.

## 2    Preliminaries

We start with some background material on conventional hypothesis testing and review linear-time kernel two-sample tests. In what follows, we will use $[d] := \{1, \ldots, d\}$ to denote the set of natural numbers up to $d \in \mathbb{N}$, $\boldsymbol{\mu} \geq \mathbf{0}$ to denote that all entries of $\boldsymbol{\mu} \in \mathbb{R}^d$ are non-negative, $e_i$ to denote the $i$-th Cartesian unit vector, and $\| \cdot \| := \| \cdot \|_2$.

**Statistical hypothesis testing.**    Let $Z$ be a random variable taking values in $\mathcal{Z} \subseteq \mathbb{R}^p$ distributed according to a distribution $P$. The goal of statistical hypothesis testing is to decide whether some *null hypothesis* $H_0$ about $P$ can be rejected in favor of an *alternative hypothesis* $H_A$ based on empirical data [2, 19]. Let $h$ be a real-valued function such that $0 < \mathbb{E}\left[h^2(Z)\right] < \infty$. In this work, we consider testing the null hypothesis $H_0 : \mathbb{E}\left[h(Z)\right] = 0$ against the one-sided alternative hypothesis $H_A : \mathbb{E}\left[h(Z)\right] > 0$ for reasons which will become clear later. To do so, we define the *test statistic* $\tau(Z_n) = \frac{1}{n}\sum_{i=1}^{n} h(z_i)$ as the empirical mean of $h$ based on a sample $Z_n := \{z_1, ..., z_n\}$ drawn i.i.d. from $P^n$. We reject $H_0$ if the observed test statistic $\hat{\tau}(Z_n)$ is *significantly* larger than what we would expect if $H_0$ was true, i.e., if $P(\tau(Z_n) < \hat{\tau}(Z_n)|H_0) > 1 - \alpha$. Here $\alpha$ is a *significance level* and controls the probability of incorrectly rejecting $H_0$ (Type-I error). For sufficiently large $n$ we can work with the asymptotic distribution of $\tau(Z_n)$, which is characterized by the Central Limit Theorem [20].

**Lemma 1.** *Let $\mu := \mathbb{E}\left[h(Z)\right]$ and $\sigma^2 := Var\left[h(Z)\right]$. Then, the test statistic converges in distribution to a Gaussian distribution, i.e., $\sqrt{n}(\tau(Z_n) - \mu) \xrightarrow{d} \mathcal{N}(0, \sigma^2)$.*

Let $\Phi$ be the CDF of the standard normal and $\Phi^{-1}$ its inverse. We define the test threshold $t_\alpha = \sqrt{n}\sigma\Phi^{-1}(1 - \alpha)$ as the $(1 - \alpha)$-quantile of the null distribution so that $P\left(\tau(Z_n) < t_\alpha|H_0\right) = 1 - \alpha$ and we reject $H_0$ simply if $\hat{\tau}(Z_n) > t_\alpha$. Besides correctly controlling the Type-I error, the test should also reject $H_0$ as often as possible when $P$ actually satisfies the alternative $H_A$. The probability of making a Type-II error is defined as $P\left(\tau(Z_n) < t_\alpha \mid H_A\right)$, i.e., the probability of failing to reject $H_0$ when it is false. A powerful test has a small Type-II error while keeping the Type-I error at $\alpha$. Since Lemma 1 holds for any $\mu$, and thus both under null and alternative hypotheses, the asymptotic probability of a Type-II error is [4]

$$P(\tau(Z_n) < t_\alpha \mid H_A) \approx \Phi\left(\Phi^{-1}(1 - \alpha) - \frac{\mu\sqrt{n}}{\sigma}\right). \tag{1}$$

Since $\Phi$ is monotonic, this probability decreases with $\mu/\sigma$, which we interpret as a signal-to-noise ratio (SNR). It is therefore desirable to find test statistics with high SNR.

**Kernel two-sample testing.**    As an example that can be expressed in the above form we present kernel two-sample tests. Given two samples $X_n$ and $Y_n$ drawn from distributions $P$ and $Q$, the two-sample test aims to decide whether $P$ and $Q$ are different, i.e., $H_0 : P = Q$ and $H_A : P \neq Q$. A popular test statistic for this problem is the maximum mean discrepancy (MMD) of Gretton et al. [3], which is defined based on a positive definite kernel function $k$ [21]: $\mathrm{MMD}^2[P, Q] = \mathbb{E}[k(x, x') + k(y, y') - k(x, y') - k(x', y)] = \mathbb{E}[h(x, x', y, y')]$, where $x, x'$ are independent draws from $P$, $y, y'$ are independent draws from $Q$, and $h(x, x', y, y') := k(x, x') + k(y, y') - k(x, y') - k(y, x')$. A minimum-variance unbiased estimator of $\mathrm{MMD}^2$ is given by a second-order $U$-statistic [20]. However, this estimator scales quadratically with the sample size, and the distribution under $H_0$ is not available in closed form. Thus it has to be simulated either via a bootstrapping approach or via a permutation of the samples. For large sample size, the computational requirements become prohibitive [3]. In this work, we assume we are in this regime. To circumvent these computational burdens, Gretton et al. [3] suggest a "linear-time" MMD estimate that scales linearly with sample size and is asymptotically normally distributed under both null and alternative hypotheses. Specifically,

let $X_{2n} = \{x_1, \ldots, x_{2n}\}$ and $Y_{2n} = \{y_1, \ldots, y_{2n}\}$, i.e., the samples are of the same (even) size. We can define $z_i := (x_i, x_{n+i}, y_i, y_{n+i})$ and $\tau(Z_n) := \frac{1}{n}\sum_{i=1}^{n} h(z_i)$ as the test statistic, which by Lemma 1 is asymptotically normally distributed. Furthermore, if the kernel $k$ is characteristic [22], it is guaranteed that $\mathrm{MMD}^2(P,Q) = 0$ if $P = Q$ and $\mathrm{MMD}^2(P,Q) > 0$ otherwise. Therefore, a one-sided test is sufficient.

Other well-known examples are goodness-of-fit tests based on the kernelized Stein discrepancy (KSD), which also has a linear time estimate [5, 6]. In our experiments, we focus on the kernel two-sample test, but point out that our theoretical treatment in Section 3 is more general and can be applied to other problems, e.g., KSD goodness-of-fit tests, but also beyond kernel methods.

## 3 Selective hypothesis tests

Statistical lore tells us *not to use the same data for learning and testing.* We now discuss whether it is indeed possible to use the same data for selecting a test statistic from a candidate set and conducting the selected test [23]. The key to controllable Type-I errors is that we need to adjust the test threshold to account for the selection event. As before, let $Z_n$ denote the data we collected. Let $T = \{\tau_i\}_{i \in \mathcal{I}}$ be a countable set of candidate test statistics that we evaluate on the data $Z_n$, and $\{t_\alpha^i\}_{i \in \mathcal{I}}$ the respective test thresholds. Assume that $\{A_i\}_{i \in \mathcal{I}}$ are disjoint *selection events* depending on $Z_n$ and that their outcomes determine which test statistic out of $T$ we apply. Thus, all the tests and events are generally dependent via $Z_n$. To define a *well-calibrated* test, we need to control the overall Type-I error, i.e., $P(\mathrm{reject}|H_0)$. Using the law of total probability, we can rewrite this in terms of the selected tests

$$P(\mathrm{reject}|H_0) = \sum_{i \in \mathcal{I}} P(\tau_i > t_\alpha^i | A_i, H_0) P(A_i | H_0). \tag{2}$$

To control the Type-I error $P(\mathrm{reject}|H_0) \leq \alpha$, it thus suffices to control $P(\tau_i > t_\alpha^i | A_i, H_0) \leq \alpha$ for each $i \in \mathcal{I}$, i.e., the test thresholds need to take into account the conditioning on the selection event $A_i$. A *naive* approach would wrongly calibrate the test such that $P(\tau_i > t_\alpha^i | H_0) \leq \alpha$, not accounting for the selection $A_i$ and thus would result in an uncontrollable Type-I error. On the other hand, this reasoning directly tells us why data splitting works. There $A_i$ is evaluated on a split of $Z_n$ that is independent of the split used to compute $\tau_i$ and hence $P(\tau_i > t_\alpha^i | A_i, H_0) = P(\tau_i > t_\alpha^i | H_0)$.

**Selecting tests with high power.** Our objective in selecting the test statistic is to maximize the power of the selected test. To this end, we start from $d \in \mathbb{N}$ different *base functions* $h_1, \ldots, h_d$. Based on observed data $Z_n = \{z_1, \ldots, z_n\} \sim P^n$, we can compute $d$ *base* test statistics $\tau_u := \tau_u(Z_n) = \frac{1}{n}\sum_{i=1}^{n} h_u(z_i)$ for $u \in [d]$. Let $\boldsymbol{\tau} := (\tau_1, \ldots, \tau_d)^\top$ and $\boldsymbol{\mu} := \mathbb{E}[\boldsymbol{h}(Z)]$, where $\boldsymbol{h}(Z) = (h_1(Z), \ldots, h_d(Z))^\top$. Asymptotically, we have $\sqrt{n}(\boldsymbol{\tau} - \boldsymbol{\mu}) \xrightarrow{d} \mathcal{N}(\boldsymbol{0}, \Sigma)$, with the variance of the asymptotic distribution given by $\Sigma = \mathrm{Cov}[\boldsymbol{h}(Z)]$.[2] Now, for any $\boldsymbol{\beta} \in \mathbb{R}^d \setminus \{\boldsymbol{0}\}$ that is independent of $\boldsymbol{\tau}$, the normalized test statistic $\tau_{\boldsymbol{\beta}} := \frac{\boldsymbol{\beta}^\top \boldsymbol{\tau}}{(\boldsymbol{\beta}^\top \Sigma \boldsymbol{\beta})^{\frac{1}{2}}}$ is asymptotically normal, i.e., $\sqrt{n}\left(\tau_{\boldsymbol{\beta}} - \frac{\boldsymbol{\beta}^\top \boldsymbol{\mu}}{(\boldsymbol{\beta}^\top \Sigma \boldsymbol{\beta})^{\frac{1}{2}}}\right) \xrightarrow{d} \mathcal{N}(0, 1)$. Following our considerations of Section 2, the test with the highest power is defined by

$$\boldsymbol{\beta}^\infty := \underset{\|\boldsymbol{\beta}\|=1}{\mathrm{argmax}} \frac{\boldsymbol{\beta}^\top \boldsymbol{\mu}}{(\boldsymbol{\beta}^\top \Sigma \boldsymbol{\beta})^{\frac{1}{2}}} = \frac{\Sigma^{-1}\boldsymbol{\mu}}{\|\Sigma^{-1}\boldsymbol{\mu}\|}, \tag{3}$$

where the constraint $\|\boldsymbol{\beta}\| = 1$ is to ensure that the solution is unique, since the objective of the maximization is a homogeneous function of order 0 in $\boldsymbol{\beta}$. The explicit form of $\boldsymbol{\beta}^\infty$ is proven in Appendix C.2. Obviously, in practice, $\boldsymbol{\mu}$ is not known, so we use an estimate of $\boldsymbol{\mu}$ to select $\boldsymbol{\beta}$. The standard strategy to do so is to split the sample $Z_n$ into two independent sets and estimate $\boldsymbol{\tau}_{\mathrm{tr}}$ and $\boldsymbol{\tau}_{\mathrm{te}}$, i.e., two independent training and test realizations [4, 8, 9, 13]. One can then choose a suitable $\boldsymbol{\beta}$ by using $\boldsymbol{\tau}_{\mathrm{tr}}$ as a proxy for $\boldsymbol{\mu}$. Then one tests with this $\boldsymbol{\beta}$ and $\boldsymbol{\tau}_{\mathrm{te}}$. However, to our knowledge, there exists no principled way to decide in which proportion to split the data, which will generally influence the power, as shown in our experimental results in Section 5.

Our approach to maximizing the utility of the observed dataset is to use it for both learning and testing. To do so, we have to derive an adjustment to the distribution of the statistic under the null, in the spirit of the selective hypothesis testing described above. We will consider three different candidate sets $T$ of test statistics, which are all constructed from the base test statistics $\boldsymbol{\tau}$. To do so, we will work with the asymptotic distribution of $\boldsymbol{\tau}$ under the null. To keep the notation concise, we include the $\sqrt{n}$ dependence into $\boldsymbol{\tau}$. Thus, we will assume $\boldsymbol{\tau} \sim \mathcal{N}(\mathbf{0}, \Sigma)$, where $\Sigma$ is known and strictly positive. We provide the generalization to singular covariance in Appendix E.

To select the test statistics, we maximize the SNR $\tau_{\boldsymbol{\beta}} = \boldsymbol{\beta}^\top \boldsymbol{\tau}/(\boldsymbol{\beta}^\top \Sigma \boldsymbol{\beta})^{\frac{1}{2}}$ and thus the test power over three different sets of candidate test statistics: 1. $T_{\text{base}} = \{\tau_{\boldsymbol{\beta}} \mid \boldsymbol{\beta} \in \{e_1, \ldots, e_d\}\}$, i.e., we directly select from the base test statistics, 2. $T_{\text{Wald}} = \{\tau_{\boldsymbol{\beta}} \mid \|\boldsymbol{\beta}\| = 1\}$, where we allow for arbitrary linear combinations, 3. $T_{\text{OST}} = \{\tau_{\boldsymbol{\beta}} \mid \Sigma \boldsymbol{\beta} \geq \mathbf{0}, \|\Sigma \boldsymbol{\beta}\| = 1\}$, where we constrain the allowed values to increase the power (see below). The rule for selecting the test statistic from these sets is simply to select the one with the highest value. To design selective hypothesis tests, we need to derive suitable selection events and the distribution of the maximum test statistic conditioned on its selection.

## 3.1 Selection from a finite candidate set

We start with $T_{\text{base}} = \{\tau_{\boldsymbol{\beta}} \mid \boldsymbol{\beta} \in \{e_1, \ldots, e_d\}\}$ and use the test statistic $\tau_{\text{base}} = \max_{\tau \in T_{\text{base}}} \tau$. Since the selection is from a countable set and the selected statistic is a projection of $\boldsymbol{\tau}$, we can use the polyhedral lemma of Lee et al. [24] to derive the conditional distributions. Therefore, we denote $u^* = \operatorname{argmax}_{u \in [d]} \frac{\tau_u}{\sigma_u}$, with $\sigma_u := (\Sigma_{uu})^{\frac{1}{2}}$, and obtain $\tau_{\text{base}} = \frac{\tau_{u^*}}{\sigma_{u^*}}$. The following corollary characterizes the conditional distribution. The proof is given in Appendix C.1.

**Corollary 1.** *Let* $\boldsymbol{\tau} \sim \mathcal{N}(\boldsymbol{\mu}, \Sigma)$, $\boldsymbol{z} := \boldsymbol{\tau} - \frac{\Sigma e_{u^*} \tau_{u^*}}{\sigma_{u^*}^2}$, $\mathcal{V}^-(\hat{\boldsymbol{z}}) = \max_{j \in [d], j \neq u^*} \frac{\sigma_{u^*} \hat{z}_j}{\sigma_u^* \sigma_j - \Sigma_{u^* j}}$, *and* $TN(\mu, \sigma^2, a, b)$ *denote a normal distribution with mean $\mu$ and variance $\sigma^2$ truncated at $a$ and $b$. Then the following statement holds:*

$$\left[ \frac{\tau_{u^*}}{\sigma_{u^*}} \,\middle|\, u^* = \operatorname*{argmax}_{u \in [d]} \frac{\tau_u}{\sigma_u}, \boldsymbol{z} = \hat{\boldsymbol{z}} \right] \stackrel{d}{=} TN\left( \frac{\mu_{u^*}}{\sigma_{u^*}}, 1, \mathcal{V}^-(\hat{\boldsymbol{z}}), \mathcal{V}^+ = \infty \right), \tag{4}$$

This scenario arises, for example, in kernel-based tests when the kernel parameters are chosen from a grid of predefined values [3, 4]. Corollary 1 allows us to test using the same set of data that was used to select the test statistic, by providing the corrected asymptotic distribution (4). The only downside is its dependence on the parameter grid. To overcome this limitation, several works have proposed to optimize for the parameters directly [4, 9–12]. Unfortunately, we cannot apply Corollary 1 directly to this scenario.

## 3.2 Learning from an uncountable candidate set

To allow for more flexible tests, in the following we consider the candidate sets $T_{\text{Wald}}$ and $T_{\text{OST}}$ that contain uncountably many tests. For these sets, we cannot directly use (2) to derive conditional tests, since the probability of selecting some given tests is 0. However, we show that it is possible in both cases to rewrite the test statistic such that we can build conditional tests based on (2). First, for $T_{\text{Wald}}$, we rewrite the entire test statistic including the maximization in closed form. Second, for $T_{\text{OST}}$ we derive suitable measurable selection events that allow us to rewrite the conditional test statistic in closed form and derive their distributions in Theorem 1.

**Wald Test.** We first allow for arbitrary linear combinations of the base test statistics $\boldsymbol{\tau}$. Therefore, define $T_{\text{Wald}} = \{\tau_{\boldsymbol{\beta}} \mid \|\boldsymbol{\beta}\| = 1\}$ and $\tau_{\text{Wald}} := \max_{\tau \in T_{\text{Wald}}} \tau$. We denote the optimal $\boldsymbol{\beta}$ for this set as $\boldsymbol{\beta}_{\text{Wald}} := \operatorname{argmax}_{\|\boldsymbol{\beta}\|=1} \frac{\boldsymbol{\beta}^\top \boldsymbol{\tau}}{(\boldsymbol{\beta}^\top \Sigma \boldsymbol{\beta})^{\frac{1}{2}}}$. This optimization problem is the same as in (3), hence $\boldsymbol{\beta}_{\text{Wald}} = \frac{\Sigma^{-1} \boldsymbol{\tau}}{\|\Sigma^{-1} \boldsymbol{\tau}\|}$, and we can rewrite the "Wald" test statistic as $\tau_{\text{Wald}} = \frac{\boldsymbol{\beta}_{\text{Wald}}^\top \boldsymbol{\tau}}{(\boldsymbol{\beta}_{\text{Wald}}^\top \Sigma \boldsymbol{\beta}_{\text{Wald}})^{\frac{1}{2}}} = (\boldsymbol{\tau}^\top \Sigma^{-1} \boldsymbol{\tau})^{\frac{1}{2}} = \|\Sigma^{-\frac{1}{2}} \boldsymbol{\tau}\|$. Note that $T_{\text{Wald}}$ contains uncountably many tests. However, instead of deriving individual conditional distributions, we can directly derive the distribution of the maximized test statistic, since $\tau_{\text{Wald}}$ can be written in closed form. In fact, under the null, we have $\Sigma^{-\frac{1}{2}} \boldsymbol{\tau} \sim \mathcal{N}(\mathbf{0}, I_d)$ and $\tau_{\text{Wald}}$ follows a chi distribution with $d$ degrees of freedom. Surprisingly, the presented approach results in the classic Wald test statistic [25], which originally was defined directly in closed form.

**One-sided test (OST).** The original Wald test was defined to optimally test $H_0 : \boldsymbol{\mu} = \mathbf{0}$ against the alternative $H_A : \boldsymbol{\mu} \neq \mathbf{0}$ [25]. Thus, it ignores the fact that we only test against the "one-sided" alternative $\boldsymbol{\mu} \geq \mathbf{0}$, which suffices since we consider linear-time estimates of the squared MMD as test statistics and their population values are non-negative. Multiplying (3) with $\Sigma$ yields $\Sigma\boldsymbol{\beta}^{\infty} = \frac{\boldsymbol{\mu}}{\|\Sigma^{-1}\boldsymbol{\mu}\|}$. Using $\boldsymbol{\mu} \geq \mathbf{0}$, we find $\Sigma\boldsymbol{\beta}^{\infty} \geq \mathbf{0}$. Thus, we have prior knowledge over the asymptotically optimal combination $\boldsymbol{\beta}^{\infty}$. To incorporate this, we a priori constrain the considered values of $\boldsymbol{\beta}$ by the condition $\Sigma\boldsymbol{\beta} \geq \mathbf{0}$. Thus we define $T_{\text{OST}} = \{\tau_{\boldsymbol{\beta}} \,|\, \Sigma\boldsymbol{\beta} \geq \mathbf{0}, \|\Sigma\boldsymbol{\beta}\| = 1\}$, where the norm constraint $\|\Sigma\boldsymbol{\beta}\| = 1$ is added to make the maximum unique. We suggest using the test statistic $\tau_{\text{OST}} := \max_{\tau \in T_{\text{OST}}} \tau$. Before we derive suitable conditional distributions for this test statistic, we rewrite it in a *canonical form*.

**Remark 1.** *Define $\boldsymbol{\alpha} := \Sigma\boldsymbol{\beta}$, $\boldsymbol{\rho} := \Sigma^{-1}\boldsymbol{\tau}$, and $\Sigma' := \Sigma^{-1}\Sigma\Sigma^{-1} = \Sigma^{-1}$. This implies $\boldsymbol{\rho} \sim \mathcal{N}(\mathbf{0}, \Sigma')$ and $\tau_{OST} := \max_{\|\Sigma\boldsymbol{\beta}\|=1, \Sigma\boldsymbol{\beta} \geq \mathbf{0}} \frac{\boldsymbol{\beta}^{\top}\boldsymbol{\tau}}{(\boldsymbol{\beta}^{\top}\Sigma\boldsymbol{\beta})^{\frac{1}{2}}} = \max_{\|\boldsymbol{\alpha}\|=1, \boldsymbol{\alpha} \geq \mathbf{0}} \frac{\boldsymbol{\alpha}^{\top}\boldsymbol{\rho}}{(\boldsymbol{\alpha}^{\top}\Sigma'\boldsymbol{\alpha})^{\frac{1}{2}}}.$*

Thus in the following, we focus on the canonical form, where the constraints are simply positivity constraints. For ease of notation, we stick with $\boldsymbol{\tau}$ and $\Sigma$ instead of $\boldsymbol{\rho}$ and $\Sigma'$. We will thus analyze the distribution of

$$\max_{\|\boldsymbol{\beta}\|=1, \boldsymbol{\beta} \geq \mathbf{0}} \frac{\boldsymbol{\beta}^{\top}\boldsymbol{\tau}}{(\boldsymbol{\beta}^{\top}\Sigma\boldsymbol{\beta})^{\frac{1}{2}}} = \frac{\boldsymbol{\beta}^{*\top}\boldsymbol{\tau}}{(\boldsymbol{\beta}^{*\top}\Sigma\boldsymbol{\beta}^{*})^{\frac{1}{2}}}, \tag{5}$$

where $\boldsymbol{\beta}^{*}(\boldsymbol{\tau}) := \text{argmax}_{\|\boldsymbol{\beta}\|=1, \boldsymbol{\beta} \geq \mathbf{0}} \frac{\boldsymbol{\beta}^{\top}\boldsymbol{\tau}}{(\boldsymbol{\beta}^{\top}\Sigma\boldsymbol{\beta})^{\frac{1}{2}}}$. We emphasize that $\boldsymbol{\beta}^{*}(\boldsymbol{\tau})$ is a random variable that is determined by $\boldsymbol{\tau}$. For conciseness, however, we will use $\boldsymbol{\beta}^{*}$ and keep the dependency implicit. We find the solution of (5) by solving an equivalent convex optimization problem, which we provide in Appendix B. We need to characterize the distribution of (5) under the null hypothesis, i.e., $\boldsymbol{\tau} \sim \mathcal{N}(\mathbf{0}, \Sigma)$. Since we are not able to give an analytic form for $\boldsymbol{\beta}^{*}$, it is hard to directly compute the distribution of $\tau_{\text{OST}}$ as we did for the Wald test. In Section 3.1 we were able to work around this by deriving the distribution conditioned on the selection of $\boldsymbol{\beta}^{*}$. In the present case, however, there are uncountably many values that $\boldsymbol{\beta}^{*}$ can take, so for some the probability is zero. Hence, the reasoning of (2) does not apply and we cannot use the PSI framework of Lee et al. [24].

Our approach to solving this is the following. Instead of directly conditioning on the explicit value of $\boldsymbol{\beta}^{*}$, we condition on the *active set*. For a given $\boldsymbol{\beta}^{*}$, we define the active set as $\mathcal{U} := \{u \,|\, \beta_u^{*} \neq 0\} \subseteq [d]$. Note that the active set is a function of $\boldsymbol{\tau}$, defined via (5). In Theorem 1 we show that given the active set, we can derive a closed-form expression for $\boldsymbol{\beta}^{*}$, and we can characterize the distribution of the test statistic conditioned on the active set. Figure 1 depicts the intuition behind Theorem 1 and Appendix A contains the full proof. In the following, let $\chi_l$ denote a chi distribution with $l$ degrees of freedom and $TN(0, 1, a, \infty)$ denote the distribution of a standard normal RV truncated from below at $a$, i.e., with CDF $F^a(x) = \frac{\Phi(x)-\Phi(a)}{1-\Phi(a)}$.

**Theorem 1.** *Let $\boldsymbol{\tau} \sim \mathcal{N}(\mathbf{0}, \Sigma)$ be a normal RV in $\mathbb{R}^d$ with positive definite covariance matrix $\Sigma$. Let $\boldsymbol{\beta}^{*}$ be defined as in (5), $\mathcal{U} := \{u \,|\, \beta_u^{*} \neq 0\}$, $l := |\mathcal{U}|$, $\boldsymbol{z} := \left(I_d - \frac{\Sigma\boldsymbol{\beta}^{*}\boldsymbol{\beta}^{*\top}}{\boldsymbol{\beta}^{*\top}\Sigma\boldsymbol{\beta}^{*}}\right)\boldsymbol{\tau}$, and $\mathcal{V}^-$ as in Corollary 1. Then, the following statements hold.*

*1.) If $l = 1$:*
$$\left[\max_{\|\boldsymbol{\beta}\|=1, \boldsymbol{\beta} \geq \mathbf{0}} \frac{\boldsymbol{\beta}^{\top}\boldsymbol{\tau}}{(\boldsymbol{\beta}^{\top}\Sigma\boldsymbol{\beta})^{\frac{1}{2}}} \,\bigg|\, \mathcal{U}, \boldsymbol{z} = \hat{\boldsymbol{z}}\right] \overset{d}{=} TN(0, 1, \mathcal{V}^-(\hat{\boldsymbol{z}}), \infty).$$

*2.) If $l \geq 2$:*
$$\left[\max_{\|\boldsymbol{\beta}\|=1, \boldsymbol{\beta} \geq \mathbf{0}} \frac{\boldsymbol{\beta}^{\top}\boldsymbol{\tau}}{(\boldsymbol{\beta}^{\top}\Sigma\boldsymbol{\beta})^{\frac{1}{2}}} \,\bigg|\, \mathcal{U}\right] \overset{d}{=} \chi_l.$$

With Theorem 1 and Remark 1, we are able to define conditional hypothesis tests with the test statistic $\tau_{\text{OST}}$. First, we transform our observation $\hat{\boldsymbol{\tau}}$ according to Remark 1 to obtain it in canonical form, i.e., $\hat{\boldsymbol{\tau}} \to \Sigma^{-1}\hat{\boldsymbol{\tau}}$ and $\Sigma \to \Sigma^{-1}$. Then we solve the optimization problem (5) to find $\boldsymbol{\beta}^{*}$. Next, we define the active set $\mathcal{U}$, by checking which entries of $\boldsymbol{\beta}^{*}$ are non-zero. Theorem 1 characterizes the distribution $\tau_{\text{OST}}$ conditioned on the selection. We can then define a test threshold $t_\alpha$ that accounts for the selection of $\mathcal{U}$, i.e.,

$$t_\alpha = \begin{cases} \Phi^{-1}\left((1-\alpha)(1-\Phi(\mathcal{V}^-)) + \Phi(\mathcal{V}^-)\right) & \text{if } |\mathcal{U}| = 1, \\ \Phi_{\chi_l}^{-1}(1-\alpha) & \text{if } |\mathcal{U}| = l \geq 2, \end{cases} \tag{6}$$

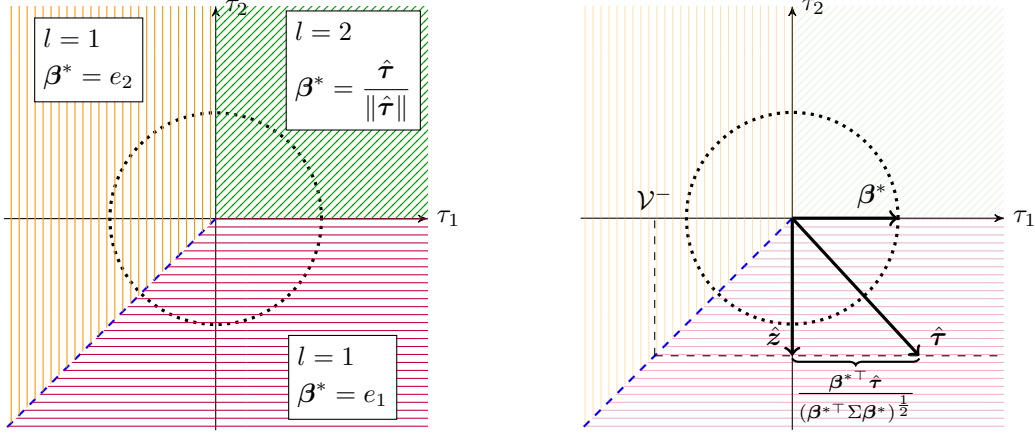

Figure 1: Geometric interpretation of Theorem 1 for $d = 2$ and unit covariance $\Sigma = I$ (denoted by the black dotted unit-circle). **Left:** If $\hat{\boldsymbol{\tau}}$ is in the positive quadrant (green), the constraints of the optimization are not active and the optimal direction is the same as for the Wald test, hence the distribution of the test statistic follows $\chi_2$. When $\hat{\boldsymbol{\tau}}$ is in the orange or purple zone, one of the constraints is active and $\boldsymbol{\beta}^*$ is a canonical unit-vector. **Right:** If $l = 1$, for example when only the first direction is active, we additionally condition on $\boldsymbol{z} = \hat{\boldsymbol{z}}$, which is independent of the value of $\boldsymbol{\beta}^{*\top}\boldsymbol{\tau}$ since $\boldsymbol{z}$ is orthogonal to $\boldsymbol{\beta}^*$. For the observed value $\hat{\boldsymbol{z}}$, we only select $\boldsymbol{\beta}^* = e_1$ if $\boldsymbol{\beta}^{*\top}\boldsymbol{\tau} \geq \mathcal{V}^-$. If this was not the case, then $\boldsymbol{\tau}$ would lie in the orange/vertically lined region and we would select $\boldsymbol{\beta}^* = e_2$. This explains the truncated behavior and is in analogy to the results of Lee et al. [24].

with $\Phi_{\chi_l}^{-1}$ being the inverse CDF of a chi distribution with $l$ degrees of freedom, which we can evaluate using standard libraries, e.g., Jones et al. [26]. We can then reject the null, if the observed value of the optimized test statistic exceeds this threshold, i.e., $\hat{\tau}_{\text{OST}} > t_\alpha$. We summarize the entire approach in Algorithm 1.

## 4 Related work

Our work is best positioned in the context of modern statistical tests with tunable hyperparameters. Gretton et al. [4] were the first to propose a kernel two-sample test that optimizes the kernel hyperparameters by maximizing the test power. This influential work has led to further development of optimized kernel-based tests [7–12]. Since any universally consistent binary classifier can be used to construct a valid two-sample test [27, 28], Kim et al. [14], Lopez-Paz and Oquab [15] used classification accuracy as a proxy to train machine learning models for two-sample tests. Kirchler et al. [17], Cai et al. [29] studied this further, and Cheng and Cloninger [16] proposed using the difference of a trained deep network's expected logit values as the test statistic for two-sample tests.

All the aforementioned "learn-then-test" approaches optimize hyperparameters (e.g., kernels, weights in a network) on a training set which is split from the full dataset. While the null distribution becomes tractable due to the independence between the optimized hyperparameters and the test set, there is a potential reduction of test power because of a smaller test set. This observation is the main motivation for our consideration of selective hypothesis tests, which allow the full dataset to be used for both training and testing by correcting for the dependency, as we discuss in Section 3.

More broadly, properly assessing the strength of potential associations that have been previously learned from the data falls under an emerging subfield of statistics known as *selective inference* [30]. A seminal work of Lee et al. [24] proposed a post-selection inference (PSI) framework to characterize the valid distribution of a post-selection estimator where model selection is performed by the Lasso [31]. The PSI framework has been applied to kernel tests, albeit in different context, for selecting the most informative features for supervised learning [32, 33], selecting a subset of features that best discriminates two samples [34], as well as selecting a model with the best fit from a list of candidate models [35]. All these applications of the PSI framework consider a finite candidate set. Our Theorem 1 can be seen as an extension of the previously known results of Lee et al. [24] to

uncountable candidate sets. To our knowledge, our work is the first to explicitly maximize test power by using the same data for selecting and testing.

Unfortunately, we cannot directly use our results to optimize tests based on complete U-statistics estimates of the MMD, which would be desirable since those estimates have lower variance than the *linear* version we use. The difficulty arises since our method requires asymptotic normality under the null, which is not the case for complete U-statistics [3]. To circumvent this problem, Yamada et al. [34] considered incomplete U-statistics [36] and Zaremba et al. [37] used a Block estimate of the MMD. Under the null, these approaches either have approximately asymptotic normal distribution [34] or require a higher sample size to reach the asymptotic normality [37]. In principle thus our approach is applicable with these methods if one is willed to assume asymptotic normality and to neglect the induced errors. Besides that, since the linear-time estimate has lowest computational cost, it should generally be used in the *large-data, constraint-computation* regime [4]. On the other hand one should consider the other approaches when the computational efforts are not the limiting factor.

Moreover, under the assumption that $\tau \sim \mathcal{N}(\boldsymbol{\mu}, \Sigma)$, similar scenarios have previously been investigated in the traditional statistical literature, but the idea of data splitting is not considered there. In particular, our construction of $\tau_{\text{Wald}}$ turned out to coincide with the test statistic suggested in Wald [25]. The one-sided version $\tau_{\text{OST}}$ also has a twin named "*chi-bar-square*" test previously considered in Kudo [38]. While their test statistic is constructed to be always non-negative, our $\tau_{\text{OST}}$ can be negative. Furthermore, they derived the distribution of the test statistic by decomposing the distribution into $2^d$ selection events, which, however, "*may represent a quite difficult problem*" [39, p. 54]. Our work circumvents this difficulty by defining a conditional test, which does not require calculating any probability of the selection events. Another difference is that our approach only defines $2^d - 1$ different active sets, by enforcing $\boldsymbol{\beta} \neq \boldsymbol{0}$. It is instructive to note that there exist other more complicate settings of "learn-then-test" scenarios in which the normality assumption may not hold [15–17, 29]. Extending our work towards these scenarios remains an open, yet promising problem to consider.

# 5 Experiments

We demonstrate the advantages of OST over data-splitting approaches and the Wald test with kernel two-sample testing problems as described in Section 2. For an extensive description of the experiments we refer to Appendix D. We consider three different datasets with different input dimensions $p$. 1. DIFF VAR ($p = 1$): $P = \mathcal{N}(0, 1)$ and $Q = \mathcal{N}(0, 1.5)$. 2. MNIST ($p = 49$): We consider downsampled 7x7 images of the MNIST dataset [40], where $P$ contains all the digits and $Q$ only uneven digits. 3. Blobs ($p = 2$): A mixture of anisotropic Gaussians where the covariance matrix of the Gaussians have different orientations for $P$ and $Q$. We denote by $k_{\text{lin}}$ the linear kernel, and $k_\sigma$ the Gaussian kernel with bandwidth $\sigma$. For each dataset we consider three different base sets of kernels $\mathcal{K}$ and choose $\tilde{\sigma}$ with the median heuristic: (a) $d = 1$: $\mathcal{K} = [k_{\tilde{\sigma}}]$, (b) $d = 2$: $\mathcal{K} = [k_{\tilde{\sigma}}, k_{\text{lin}}]$, (c) $d = 6$: $\mathcal{K} = [k_{0.25\tilde{\sigma}}, k_{0.5\tilde{\sigma}}, k_{\tilde{\sigma}}, k_{2\tilde{\sigma}}, k_{4\tilde{\sigma}}, k_{\text{lin}}]$. From the base set of kernels we estimate the base set of test statistics using the linear-time MMD estimates. We compare four different approaches: i) OST, ii) WALD, iii) SPLIT: Data splitting similar to the approach in Gretton et al. [4], but with the same constraints as OST. SPLIT0.1 denotes that 10% of the data are used for learning $\boldsymbol{\beta}^*$ and 90% are used for testing, iv) NAIVE: Similar to splitting but all the data is used for learning and testing without correcting for the dependency. The NAIVE approach is not a well-calibrated test. For all the setups we estimate the Type-II error for various sample sizes at a level $\alpha = 0.05$. Error rates are estimated over 5000 independent trials and the results are shown in Figure 2. In Appendix D.1, we also investigate the Type-I error and show that all methods except for NAIVE correctly control the Type-I error at a rate $\alpha$. Note that all of the methods scale with $\mathcal{O}(n)$ and the difference in computational cost are negligible.

The experimental results in Figure 2 support the main claims of this paper. First, comparing OST with SPLIT, we conclude that using all the data in an integrated approach is always better (or equally good) than any data splitting approach. Second, comparing OST to WALD, we conclude that adding a priori information ($\boldsymbol{\mu} \geq \boldsymbol{0}$) to reduce the class of considered tests in a sensible way leads to higher (or equally high) test power. Another interesting observation is in the results of the data-splitting approach. Looking at the DIFF VAR experiment, in the leftmost plot, we can see that the errors are monotonically increasing with the portion of data used to select the test. Since there is only one test, the more data we use to select the test, the higher the error (less data remains for testing). In the

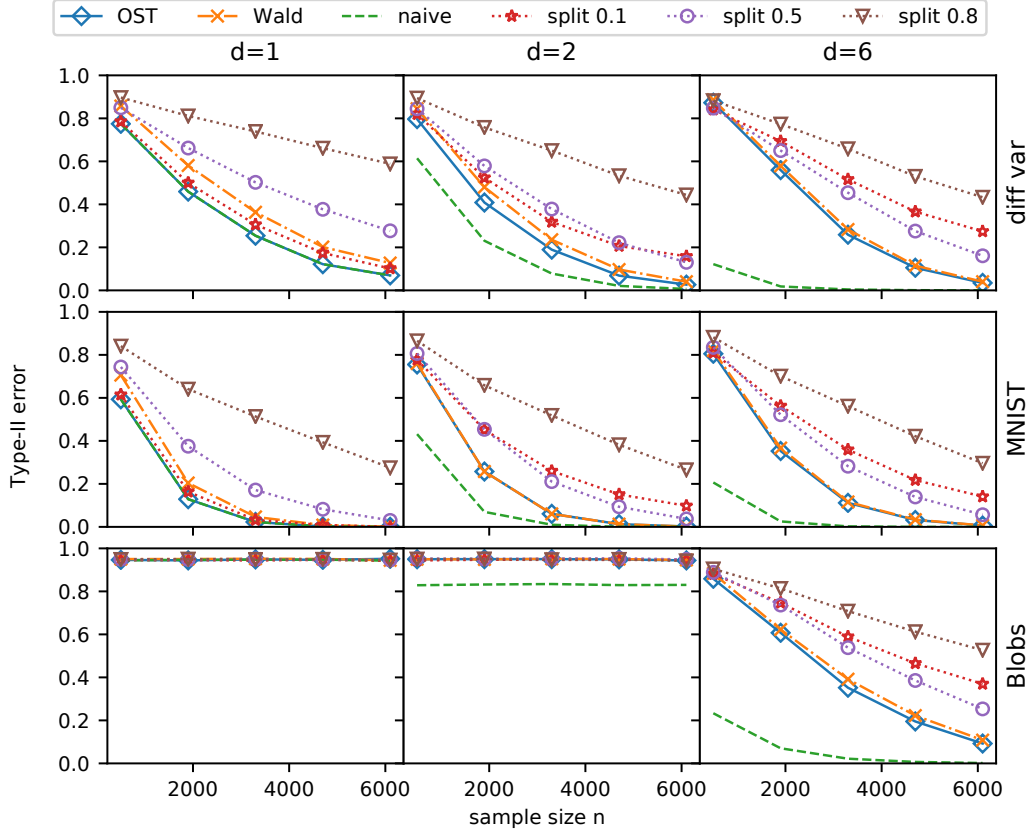

Figure 2: Type-II errors obtained from different experiments. The rows (columns) correspond to different datasets (sets of base kernels). For all considered cases, OST outperforms all the (well-calibrated) competing methods, i.e., SPLIT and WALD.

middle plot, selection becomes important. Hence, we can see that the gap in performance between all data-splitting approach reduces. However, the order is still consistent with the previous plot. Interestingly, in the rightmost plot, learning becomes even more important. Now, the order changes. If we use too little data for learning the test (SPLIT0.1), the error is high. However, if we use too much data for learning the test (SPLIT0.8), the error will be high as well. That is, there is a trade-off in how much data one should use for selecting the test, and for conducting the test. The optimal proportion depends on the problem and can thus in general not be determined a priori.

In the Appendix D.3 we also compare $\tau_{\text{base}}$ to a selection of a base test via the data-splitting approach. Here, SPLIT0.1 consistently performs better than the other split approaches, which is plausible, since the class of considered tests $T_{\text{base}}$ is quite small. SPLIT0.1 can even be better than $\tau_{\text{base}}$, see discussion in Appendix D.3.

In Figure 3, we additionally consider a constructed 1-D dataset where the distributions share the first three moments and all uneven moments vanish (Figure 7 in the appendix). We compare the results for different sets of $d \in [5]$ base kernels $\mathcal{K} = [k_{\text{pol}}^1, \ldots, k_{\text{pol}}^d]$, where $k_{\text{pol}}^u(x, y) = (x \cdot y)^u$ denotes the homogeneous polynomial kernel of order $u$. By construction, $k_{\text{pol}}^u$ does not contain any information about the difference of $P$ and $Q$, for $u \neq 4$. Thus, for $d \leq 3$ the well-calibrated methods have a Type-II error of $1 - \alpha$. Only the NAIVE approach already overfits to the noise. Adding the fourth order polynomial adds helpful information and all the methods improve performance. However, adding the fifth order, which again only contains noise, leads to an increased error rate. We interpret this as bias-variance tradeoff that should be considered in the construction of the base set $\mathcal{K}$.

**Algorithm 1** One-Sided Test (OST)

---

**input** $\Sigma, \hat{\boldsymbol{\tau}} = \sqrt{n}\widehat{\mathrm{MMD}}^2(P,Q), \alpha$

$\quad \hat{\boldsymbol{\tau}} = \Sigma^{-1}\hat{\boldsymbol{\tau}}$ {Apply Remark 1}

$\quad \Sigma = \Sigma^{-1}$ {Apply Remark 1}

$\quad \boldsymbol{\beta}^* = \mathrm{argmax}_{\|\boldsymbol{\beta}\|=1, \boldsymbol{\beta}\geq\mathbf{0}} \dfrac{\boldsymbol{\beta}^\top\hat{\boldsymbol{\tau}}}{(\boldsymbol{\beta}^\top\Sigma\boldsymbol{\beta})^{\frac{1}{2}}}$

$\quad \mathcal{U} = \{u | u \in [d], \beta_u^* > 0\}$

$\quad \hat{\boldsymbol{z}} = \hat{\boldsymbol{\tau}} - \Sigma\boldsymbol{\beta}^* \dfrac{\boldsymbol{\beta}^{*\top}\hat{\boldsymbol{\tau}}}{\boldsymbol{\beta}^{*\top}\Sigma\boldsymbol{\beta}^*}$

$\quad l = |\mathcal{U}|$

$\quad$**if** $l \geq 2$ **then**

$\quad\quad t_\alpha = \Phi_{\chi_l}^{-1}(1-\alpha)$

$\quad$**if** $l = 1$ **then**

$\quad\quad \mathcal{V}^- = \max_{u\notin\mathcal{U}} \dfrac{\hat{z}_u(\boldsymbol{\beta}^{*\top}\Sigma\boldsymbol{\beta}^*)^{\frac{1}{2}}}{\Sigma_{uu}^{\frac{1}{2}}(\boldsymbol{\beta}^{*\top}\Sigma\boldsymbol{\beta}^*)^{\frac{1}{2}} - (\Sigma\boldsymbol{\beta}^*)_u}$

$\quad\quad t_\alpha = \Phi^{-1}\left((1-\alpha)(1-\Phi(\mathcal{V}^-)) + \Phi(\mathcal{V}^-)\right)$

$\quad$**if** $t_\alpha < \dfrac{\boldsymbol{\beta}^{*\top}\hat{\boldsymbol{\tau}}}{(\boldsymbol{\beta}^{*\top}\Sigma\boldsymbol{\beta}^*)^{\frac{1}{2}}}$ **then**

$\quad\quad$ Reject $H_0$

---

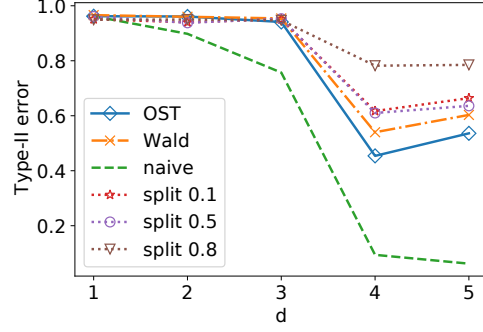

Figure 3: Type-II errors when the first $d$ polynomial kernels are used for a two-sample test with symmetric distributions with the equal covariance (Figure 7 in the appendix). OST outperforms all the (well-calibrated) competitors.

In Appendix D.2 we compare how the constraints $\boldsymbol{\beta} \geq \mathbf{0}$, as suggested in Gretton et al. [4], work in comparison to the OST approach. We find that while the constraints $\Sigma\boldsymbol{\beta} \geq \mathbf{0}$ lead to consistently higher power than the Wald test, the simple positivity constraints can lead to both, better or worse power depending on the problem. We thus recommend using the OST.

# 6  Conclusion

Previous work used data splitting to exclude dependencies when optimizing a hypothesis test. This work is the first step towards using all the data for learning and testing. Our approach uses asymptotic joint normality of a predefined set of test statistics to derive the conditional null distributions in closed form. We investigated the example of kernel two-sample tests, where we use linear-time MMD estimates of multiple kernels as a base set of test statistics. We experimentally verified that an integrated approach outperforms the existing data-splitting approach of Gretton et al. [4]. Thus data splitting, although theoretically easy to justify, does not efficiently use the data. Further, we experimentally showed that a one-sided test (OST), using prior information about the alternative hypothesis, leads to an increase in test power compared to the more general Wald test. Since the estimates of the base test statistics are linear in the sample size and the null distributions are derived analytically, the whole procedure is computationally cheap. However, it is an open question whether and how this work can be generalized to problems where the class of candidate tests is not directly constructed from a base set of jointly normal test statistics.

## Broader impact

Hypothesis testing and valid inference after model selection are fundamental problems in statistics, which have recently attracted increasing attention also in machine learning. Kernel tests such as MMD are not only used for statistical testing, but also to design algorithms for deep learning and GANs [41, 42]. The question of how to select the test statistic naturally arises in kernel-based tests because of the kernel choice problem. Our work shows that it is possible to overcome the need of (wasteful and often heuristic) data splitting when designing hypothesis tests with feasible null distribution. Since this comes without relevant increase in computational resources we expect the proposed method to replace the data splitting approach in applications that fit the framework considered in this work. Theorem 1 is also applicable beyond hypothesis testing and extends the previously known PSI framework proposed by Lee et al. [24].

## Acknowledgments and Disclosure of Funding

The authors thank Arthur Gretton, Will Fithian, and Kenji Fukumizu for helpful discussion. JMK thanks Simon Buchholz for helpful discussions and pointing out a simplification of Lemma 2.

## Footnotes

[2] In practice, we work with an estimate $\hat{\Sigma}$ of the covariance obtained from $Z_n$, which is justified since $\sqrt{n}\hat{\Sigma}^{-\frac{1}{2}}(\boldsymbol{\tau} - \boldsymbol{\mu}) \xrightarrow{d} \mathcal{N}(\boldsymbol{0}, I_d)$ for consistent estimates of the covariance.

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
