[Supplementary Material]

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

# A Proof of Theorem 1

In this section we prove the main theorem. The outline of the proof is as follows: We first characterize the "selection event", i.e., we characterize under which conditions each active set $\mathcal{U}$ is selected. This is done with Lemmas 2 and 3. For the case $l = 1$ we then show that the PSI framework of Lee et al. [24] can be applied and we recover the result of Corollary 1. It is not surprising, that for the case $l = 1$ the PSI framework works, since $\mathcal{U}$ corresponds to a single fixed $\boldsymbol{\beta}^*$ and the probability of selecting it is greater than 0. For the case $l \geq 2$, we show, that the considered test statistic essentially takes the same form as the Wald test but only on the active dimensions. Thus it follows a $\chi_l$ distribution. This distribution does not change even if we explicitly condition on the selection of $\mathcal{U}$. This is because the randomness that determines which active set is selected is independent of the value of the selected test statistic. Before we start with the proof we collect some notation we introduce for the proof.
**Notation:**

- The objective of the optimization $f(\boldsymbol{\beta}) := \frac{\boldsymbol{\beta}^\top \boldsymbol{\tau}}{(\boldsymbol{\beta}^\top \Sigma \boldsymbol{\beta})^{\frac{1}{2}}}$.

- Projector onto the active subspace (leaving the dependency on $\mathcal{U}$ implicit):

$$\Pi := \sum_{u \in \mathcal{U}} e_u e_u^\top,$$

  where $e_u$ denotes the $u$-th Cartesian unit vector in $\mathbb{R}^d$.

- $\boldsymbol{z} := \left( I_d - \frac{\Sigma \boldsymbol{\beta}^* \boldsymbol{\beta}^{*\top}}{\boldsymbol{\beta}^{*\top} \Sigma \boldsymbol{\beta}^*} \right) \boldsymbol{\tau} = \boldsymbol{\tau} - \Sigma \boldsymbol{\beta}^* \frac{\boldsymbol{\beta}^{*\top} \boldsymbol{\tau}}{\boldsymbol{\beta}^{*\top} \Sigma \boldsymbol{\beta}^*}$.

- $\bar{\Sigma}$ denotes the pseudoinverse of $\Pi \Sigma \Pi$.

As a first step, we need to characterize which values of $\boldsymbol{\tau}$ correspond to which active set $\mathcal{U}$. This is done with Lemma 2, which we prove separately in A.1.

**Lemma 2.** *Let* $\mathcal{U} := \{u \mid \beta_u^* \neq 0\}$. *Then,*

$$\boldsymbol{\beta}^* = \underset{\|\boldsymbol{\beta}\|=1, \boldsymbol{\beta} \geq \mathbf{0}}{\operatorname{argmax}} \frac{\boldsymbol{\beta}^\top \boldsymbol{\tau}}{(\boldsymbol{\beta}^\top \Sigma \boldsymbol{\beta})^{\frac{1}{2}}}$$

*if and only if all of the following conditions hold:*

1. $\left. \frac{\partial}{\partial \beta_u} \frac{\boldsymbol{\beta}^\top \boldsymbol{\tau}}{(\boldsymbol{\beta}^\top \Sigma \boldsymbol{\beta})^{\frac{1}{2}}} \right|_{\boldsymbol{\beta}=\boldsymbol{\beta}^*} \begin{cases} \leq 0 \text{ if } u \notin \mathcal{U} & (a), \\ = 0 \text{ if } u \in \mathcal{U} & (b), \end{cases}$

2. $\frac{\boldsymbol{\beta}^{*\top} \boldsymbol{\tau}}{(\boldsymbol{\beta}^{*\top} \Sigma \boldsymbol{\beta}^*)^{\frac{1}{2}}} \geq \frac{\tau_u}{\sqrt{\Sigma_{uu}}}, \qquad \forall u \notin \mathcal{U},$

3. $\beta_u^* = 0 \quad \forall u \notin \mathcal{U} \quad (a),$
   $\beta_u^* > 0 \quad \forall u \in \mathcal{U} \quad (b),$
   $\|\boldsymbol{\beta}^*\| = 1 \quad (c).$

Intuitively, Condition1(b) ensures that $\boldsymbol{\beta}^*$ is a local maximum of the objective function for the active dimensions. Condition 1(a) ensures that if $u \notin \mathcal{U}$, increasing $\beta_u^*$ does not improve the SNR. Condition 2 is harder to interpret, but is needed in cases where all entries of $\boldsymbol{\tau}$ are negative. Condition 3 enforces that $\boldsymbol{\beta}^*$ lies in the feasible set of (5).

Note that $\boldsymbol{\beta}^{*\top} \boldsymbol{\tau}$ is essentially a one-dimensional RV. We define another random variable

$$\boldsymbol{z} := \left( I_d - \frac{\Sigma \boldsymbol{\beta}^* \boldsymbol{\beta}^{*\top}}{\boldsymbol{\beta}^{*\top} \Sigma \boldsymbol{\beta}^*} \right) \boldsymbol{\tau} = \boldsymbol{\tau} - \Sigma \boldsymbol{\beta}^* \frac{\boldsymbol{\beta}^{*\top} \boldsymbol{\tau}}{\boldsymbol{\beta}^{*\top} \Sigma \boldsymbol{\beta}^*}. \tag{7}$$

In Appendix A.2, we show that $\boldsymbol{z}$ is closely related to the partial derivatives of the objective function and we have

$$\left. \frac{\partial}{\partial \beta_u} \frac{\boldsymbol{\beta}^\top \boldsymbol{\tau}}{(\boldsymbol{\beta}^\top \Sigma \boldsymbol{\beta})^{\frac{1}{2}}} \right|_{\boldsymbol{\beta}=\boldsymbol{\beta}^*} = \frac{\boldsymbol{z}}{(\boldsymbol{\beta}^{*\top} \Sigma \boldsymbol{\beta}^*)^{\frac{1}{2}}}. \tag{8}$$

We can then rewrite the conditions of Lemma 2 as follows.

**Lemma 3.** *The conditions of Lemma 2 are equivalent to*

1. $\begin{cases} z_u \leq 0 & \forall u \notin \mathcal{U} & (a), \\ z_u = 0 & \forall u \in \mathcal{U} & (b), \end{cases}$

2. $\dfrac{\boldsymbol{\beta}^{*\top}\boldsymbol{\tau}}{(\boldsymbol{\beta}^{*\top}\Sigma\boldsymbol{\beta}^*)^{\frac{1}{2}}} \geq \mathcal{V}^-(\boldsymbol{z})$, with

$$\mathcal{V}^-(\boldsymbol{z}) := \max_{u \notin \mathcal{U}} \frac{z_u(\boldsymbol{\beta}^{*\top}\Sigma\boldsymbol{\beta}^*)^{\frac{1}{2}}}{\Sigma_{uu}^{\frac{1}{2}}(\boldsymbol{\beta}^{*\top}\Sigma\boldsymbol{\beta}^*)^{\frac{1}{2}} - (\Sigma\boldsymbol{\beta}^*)_u},$$

3. $\begin{aligned} \boldsymbol{\beta}_u^* &= 0 & \forall u \notin \mathcal{U} & (a), \\ \boldsymbol{\beta}_u^* &> 0 & \forall u \in \mathcal{U} & (b), \\ \|\boldsymbol{\beta}^*\| &= 1 & & (c). \end{aligned}$

*Proof of Lemma 3.* Condition 1 directly follows from (8). The second condition follows by inserting the definition of $\boldsymbol{z}$

$$\frac{\boldsymbol{\beta}^{*\top}\boldsymbol{\tau}}{(\boldsymbol{\beta}^{*\top}\Sigma\boldsymbol{\beta}^*)^{\frac{1}{2}}} \geq \frac{\tau_u}{\sqrt{\Sigma_{uu}}}$$

$$\Leftrightarrow \frac{\boldsymbol{\beta}^{*\top}\boldsymbol{\tau}}{(\boldsymbol{\beta}^{*\top}\Sigma\boldsymbol{\beta}^*)^{\frac{1}{2}}} \geq \frac{z_u}{\sqrt{\Sigma_{uu}}} + e_u^\top \Sigma \boldsymbol{\beta}^* \frac{\boldsymbol{\beta}^{*\top}\boldsymbol{\tau}}{\boldsymbol{\beta}^{*\top}\Sigma\boldsymbol{\beta}^* \sqrt{\Sigma_{uu}}}$$

$$\Leftrightarrow \frac{\boldsymbol{\beta}^{*\top}\boldsymbol{\tau}}{(\boldsymbol{\beta}^{*\top}\Sigma\boldsymbol{\beta}^*)^{\frac{1}{2}}} \left(1 - \frac{e_u^\top \Sigma \boldsymbol{\beta}^*}{(\boldsymbol{\beta}^{*\top}\Sigma\boldsymbol{\beta}^*)^{\frac{1}{2}} \sqrt{\Sigma_{uu}}}\right) \geq \frac{z_u}{\sqrt{\Sigma_{uu}}}$$

$$\Leftrightarrow \frac{\boldsymbol{\beta}^{*\top}\boldsymbol{\tau}}{(\boldsymbol{\beta}^{*\top}\Sigma\boldsymbol{\beta}^*)^{\frac{1}{2}}} \left((\boldsymbol{\beta}^{*\top}\Sigma\boldsymbol{\beta}^*)^{\frac{1}{2}} \sqrt{\Sigma_{uu}} - e_u^\top \Sigma \boldsymbol{\beta}^*\right) \geq z_u(\boldsymbol{\beta}^{*\top}\Sigma\boldsymbol{\beta}^*)^{\frac{1}{2}}$$

$$\Leftrightarrow \frac{\boldsymbol{\beta}^{*\top}\boldsymbol{\tau}}{(\boldsymbol{\beta}^{*\top}\Sigma\boldsymbol{\beta}^*)^{\frac{1}{2}}} \geq \frac{z_u(\boldsymbol{\beta}^{*\top}\Sigma\boldsymbol{\beta}^*)^{\frac{1}{2}}}{\left((\boldsymbol{\beta}^{*\top}\Sigma\boldsymbol{\beta}^*)^{\frac{1}{2}} \sqrt{\Sigma_{uu}} - e_u^\top \Sigma \boldsymbol{\beta}^*\right)},$$

where we used $\Sigma_{uu}^{\frac{1}{2}}(\boldsymbol{\beta}^{*\top}\Sigma\boldsymbol{\beta}^*)^{\frac{1}{2}} - (\Sigma\boldsymbol{\beta}^*)_u > 0$, which holds since $\Sigma$ is positive and we only consider $u$ such that $e_u \neq \boldsymbol{\beta}^*$. $\qquad\square$

Note that $\mathcal{V}^-(\boldsymbol{z})$ is always non-positive by Condition 1 and the positivity of $\Sigma$. With the above two lemmas we are able to prove Theorem 1.

*Proof of Theorem 1.* We prove the two cases $l = 1$ and $l \geq 2$ separately.

1.): Let $u^* \in [d]$ such that $\mathcal{U} = \{u^*\}$. In this case, by Condition 3, $\boldsymbol{\beta}^* = e_{u^*}$. We shall now see how Lemma 3 constrains the distribution of $\tau_{u^*}$. For Condition 1(b), we have $z_{u^*} = 0$ by the definition of $\boldsymbol{z}$. So there only remain the constraints 1(a) and 2. Using the definition (7) of $\boldsymbol{z}$, we can rewrite 1(a) as

$$\left(\left(I_d - \Sigma e_{u^*}\frac{e_{u^*}^\top}{\Sigma_{u^*u^*}}\right)\boldsymbol{\tau}\right)_u \leq 0 \quad \forall u \notin \mathcal{U} \iff A^{[1(b)]}\boldsymbol{\tau} \leq 0,$$

where $A^{[1(b)]}$ is the matrix $\left(I_d - \Sigma e_{u^*}\frac{e_{u^*}^\top}{\Sigma_{u^*u^*}}\right)$ and we used that its $u$-th row contains only zeros.

Note that Condition 2 is the same as used in Section 3.1. Thus we can define the matrix $A^{[2]}$ as we do in the proof of Corollary 1. We have now all the remaining constraints as linear inequalities of $\boldsymbol{\tau}$ and thus we can find the conditional distribution by applying Theorem 2. Defining $\boldsymbol{\eta} = \frac{e_{u^*}}{(\boldsymbol{\beta}^*T\Sigma\boldsymbol{\beta}^*)^{\frac{1}{2}}}$

and $\boldsymbol{c} := \Sigma\boldsymbol{\eta}\left(\boldsymbol{\eta}^\top\Sigma\boldsymbol{\eta}\right)^{-1}$, we get $A^{[1(b)]}\boldsymbol{c} = \boldsymbol{0}$. Note that whenever $(A\boldsymbol{c})_j = 0$, the constraint does not change anything in Theorem 2. Thus the result follows by using $A = A^{[2]}$ and application of Theorem 2.

An alternative proof can be done by noting that $\boldsymbol{z}$ is independent of $\frac{\boldsymbol{\beta}^{*\top}\boldsymbol{\tau}}{(\boldsymbol{\beta}^{*\top}\Sigma\boldsymbol{\beta}^*)^{\frac{1}{2}}}$ if we consider $\boldsymbol{\beta}^* = e_{u^*}$ as fixed. Thus, the fulfillment of Condition 1b) is independent of $\frac{\boldsymbol{\beta}^{*\top}\boldsymbol{\tau}}{(\boldsymbol{\beta}^{*\top}\Sigma\boldsymbol{\beta}^*)^{\frac{1}{2}}}$. Since the

unconditional distribution of $\frac{\boldsymbol{\beta^*}^\top \boldsymbol{\tau}}{(\boldsymbol{\beta^*}^\top \Sigma \boldsymbol{\beta^*})^{\frac{1}{2}}}$ follows a standard normal, adding Condition 2 results in a truncated normal.

2.) Next, we consider the case $|\mathcal{U}| \geq 2$. Again we will be considering the conditions as stated in Lemma 3. As we state in (15), we have $\boldsymbol{\beta^*}^\top \boldsymbol{\tau} \geq 0$ and thus Condition 2 is fulfilled, since $\mathcal{V}^-$ is always non-positive. Thus, we can neglect Condition 2. Our first step will be to find a closed form function $h_\mathcal{U}$ such that $\boldsymbol{\beta^*} = h_\mathcal{U}(\boldsymbol{\tau})$ (this function will only hold true if $\mathcal{U}$ is actually the active set). Defining the projector onto the active subspace $\Pi := \sum_{u \in \mathcal{U}} e_u e_u^\top$, by Condition 3(a) we have $\boldsymbol{\beta^*} = \Pi \boldsymbol{\beta^*}$. Using (7), we can rewrite Condition 1(b) as

$$\Pi \boldsymbol{z} = \boldsymbol{0} \overset{(7)}{\Leftrightarrow} \Pi \boldsymbol{\tau} = \Pi \Sigma \boldsymbol{\beta^*} \frac{\boldsymbol{\beta^*}^\top \boldsymbol{\tau}}{\boldsymbol{\beta^*}^\top \Sigma \boldsymbol{\beta^*}} \overset{3(a)}{\Leftrightarrow} \Pi \boldsymbol{\tau} = \Pi \Sigma \Pi \boldsymbol{\beta^*} \frac{\boldsymbol{\beta^*}^\top \boldsymbol{\tau}}{\boldsymbol{\beta^*}^\top \Sigma \boldsymbol{\beta^*}}. \tag{9}$$

This defines a system of $l$ non-trivial equations and by Condition 3, $\boldsymbol{\beta^*}$ has $l$ free parameters. We define $\bar{\Sigma}$ as the pseudoinverse of $\Pi \Sigma \Pi$.[3] For the pseudoinverse it is easy to show $\bar{\Sigma} = \Pi \bar{\Sigma} = \bar{\Sigma} \Pi$. Since $\Sigma$ has full rank, a possible solution of (9) necessarily has to be of the form $\boldsymbol{\beta^*} = c \cdot \bar{\Sigma} \boldsymbol{\tau}$ for some $c \in \mathbb{R}$. Plugging this into (9), we get $c = \frac{\boldsymbol{\beta^*}^\top \Sigma \boldsymbol{\beta^*}}{\boldsymbol{\beta^*}^\top \boldsymbol{\tau}}$. Using (15) we get $0 \leq \frac{\boldsymbol{\beta^*}^\top \boldsymbol{\tau}}{(\boldsymbol{\beta^*}^\top \Sigma \boldsymbol{\beta^*})^{\frac{1}{2}}} = \frac{1}{c}$. Hence, $c \geq 0$. Using $\|\boldsymbol{\beta^*}\| = 1$ we get $c = \frac{1}{\|\bar{\Sigma}\boldsymbol{\tau}\|}$. Thus, given that the active set is $\mathcal{U}$, we found a closed-form solution for $\boldsymbol{\beta^*}$ as a function of $\boldsymbol{\tau}$, i.e.,

$$\boldsymbol{\beta^*} = h_\mathcal{U}(\boldsymbol{\tau}) := \frac{\bar{\Sigma} \boldsymbol{\tau}}{\|\bar{\Sigma} \boldsymbol{\tau}\|}. \tag{10}$$

Note that so far we did not use Condition 3(b), so this formula itself does not ensure the positivity of $\boldsymbol{\beta^*}$.

Replacing $\boldsymbol{\beta^*}$ in the definition (7) of $\boldsymbol{z}$ with its closed form, the constant $c$ cancels, and we get

$$\boldsymbol{z} = \boldsymbol{\tau} - \Sigma \bar{\Sigma} \boldsymbol{\tau}.$$

Note that $\bar{\Sigma} \Pi \Sigma \Pi \bar{\Sigma} = \bar{\Sigma}$ and $(\Sigma \bar{\Sigma})_{uu'} = \delta_{uu'}$ if $u, u' \in \mathcal{U}$. This implies that $z_u = 0$ if $u \in \mathcal{U}$ and thus also $\boldsymbol{z}^\top \bar{\Sigma} \boldsymbol{\tau} = 0$.

Let us now define $\tilde{X} := (\bar{\Sigma})^{\frac{1}{2}} \boldsymbol{\tau}$, resulting in $\tilde{X}_u = 0$ for all $u \notin \mathcal{U}$. Since $\tilde{X}$ and $\boldsymbol{z}$ are both linear transformations of $\boldsymbol{\tau}$ they are jointly normally distributed. In Appendix A.3 we show that $\tilde{X}$ and $\boldsymbol{z}$ are uncorrelated. This, together with the joint normality, implies that they are independent, i.e.,

$$\tilde{X} \perp\!\!\!\perp \boldsymbol{z}. \tag{11}$$

Further the non-zero coordinates of $\tilde{X}$ are jointly distributed according to a $l$-dimensional standard normal distribution. Hence, its euclidean norm follows a chi-distribution

$$\|\tilde{X}\|_2 \sim \chi_l. \tag{12}$$

Let us summarize how we used all the conditions of Lemma 3 and finish the proof. We used 1(b), 3(a), and 3(c) to show (10). We thus still need to condition on 1(a), and 3(b). Conditioning on 1(a) can be done using the independence of $\boldsymbol{z}$ and $\tilde{X}$. To condition on 3(b), we rewrite it in terms of $\tilde{X}$, i.e., for all $u \in \mathcal{U}$ we have

$$\boldsymbol{\beta}_u^* > 0 \Leftrightarrow (\bar{\Sigma}\boldsymbol{\tau})_u \Leftrightarrow \left((\bar{\Sigma})^{\frac{1}{2}} \tilde{X}\right)_u > 0 \Leftrightarrow \left((\bar{\Sigma})^{\frac{1}{2}} \frac{\tilde{X}}{\|\tilde{X}\|}\right)_u > 0.$$

Thus it only depends on the direction of $\tilde{X}$. Since the non-trivial entries of $\tilde{X}$ follow a standard normal, the direction of $\tilde{X}$ is independent of its norm, i.e.,

$$\|\tilde{X}\|_2 \perp\!\!\!\perp \frac{\tilde{X}}{\|\tilde{X}\|_2}. \tag{13}$$

Figure 4: Numerical verification of Theorem 1. For the histogram, we generate a random covariance matrix $\Sigma \in \mathbb{R}^{4\times 4}$ and sample $\boldsymbol{\tau} \sim \mathcal{N}(\mathbf{0}, \Sigma)$. We solve (5) and only accept the samples for which the active set is $\mathcal{U} = \{1, 2\}$. The orange line is the theoretical distribution according to Theorem 1, which is given by a chi distribution with two degrees of freedom. For the specific example the acceptance rate is $P(\mathcal{U} = \{1, 2\}) \approx 4\%$.

In the end we get

$$
\left[\frac{\boldsymbol{\beta}^*\boldsymbol{\tau}}{(\boldsymbol{\beta}^*\Sigma\boldsymbol{\beta}^*)^{\frac{1}{2}}}\middle|\text{Conditions } 1, 2, 3\right] \overset{(10)}{\to} \overset{d}{=} \left[\frac{\boldsymbol{\tau}^\top \bar{\Sigma}\boldsymbol{\tau}}{(\boldsymbol{\tau}\bar{\Sigma}\boldsymbol{\tau})^{\frac{1}{2}}}\middle|\text{Conditions } 1(a), 3(b)\right]
$$

$$
\overset{d}{=} \left[\|\tilde{X}\|_2\middle| \begin{cases} z_u \le 0 & \forall u \notin \mathcal{U}, \\ \left((\bar{\Sigma})^{\frac{1}{2}}\frac{\tilde{X}}{\|\tilde{X}\|}\right)_u > 0 & \forall u \in \mathcal{U} \end{cases}\right] \overset{(11)}{\underset{(13)}{\to}} \overset{d}{=} \left[\|\tilde{X}\|_2\right] \overset{d}{=} \chi_l.
$$

$\square$

## A.1   Proof of Lemma 2

*Proof of Lemma 2.* Since the objective is a homogeneous function of order zero in $\boldsymbol{\beta}$, we can make the proof by considering the optimization without the constraint $\|\boldsymbol{\beta}\| = 1$.

The necessity of the conditions is trivial to show. We thus only show the sufficiency. The fourth condition ensures that $\boldsymbol{\beta}^*$ is in the feasible set. For the other conditions, assume there exists $\boldsymbol{\xi} \in \mathbb{R}^d$ such that $\xi_u \ge 0$ for all $u \in [d]$ and $\frac{\boldsymbol{\xi}^\top \boldsymbol{\tau}}{(\boldsymbol{\xi}^\top \Sigma \boldsymbol{\xi})^{\frac{1}{2}}} > \frac{\boldsymbol{\beta}^{*\top}\boldsymbol{\tau}}{(\boldsymbol{\beta}^{*\top}\Sigma\boldsymbol{\beta}^*)^{\frac{1}{2}}}$. In the following we show that this implies that at least one of the conditions above is violated, and hence the conditions are sufficient. We separate two cases, $i)$ where $\boldsymbol{\beta}^{*\top}\boldsymbol{\tau} \ge 0$, and $ii)$ $\boldsymbol{\beta}^{*\top}\boldsymbol{\tau} < 0$.

i) Assume $\boldsymbol{\beta^*}^\top \boldsymbol{\tau} \geq 0$. We have

$$\boldsymbol{\xi}^\top \nabla_{\boldsymbol{\beta}} \left. \frac{\boldsymbol{\beta}^\top \boldsymbol{\tau}}{(\boldsymbol{\beta}^\top \Sigma \boldsymbol{\beta})^{\frac{1}{2}}} \right|_{\boldsymbol{\beta}=\boldsymbol{\beta^*}}$$

$$= \sum_{u \in [d]} \xi_u \left. \frac{\partial}{\partial \beta_u} \frac{\boldsymbol{\beta}^\top \boldsymbol{\tau}}{(\boldsymbol{\beta}^\top \Sigma \boldsymbol{\beta})^{\frac{1}{2}}} \right|_{\boldsymbol{\beta}=\boldsymbol{\beta^*}}$$

$$= \frac{\boldsymbol{\xi}^\top \boldsymbol{\tau}}{(\boldsymbol{\beta^*}^\top \Sigma \boldsymbol{\beta^*})^{\frac{1}{2}}} - \frac{\boldsymbol{\beta^*}^\top \boldsymbol{\tau}}{(\boldsymbol{\beta^*}^\top \Sigma \boldsymbol{\beta^*})^{\frac{3}{2}}} \boldsymbol{\xi}^\top \Sigma \boldsymbol{\beta^*}$$

$$= \frac{(\boldsymbol{\xi}^\top \Sigma \boldsymbol{\xi})^{\frac{1}{2}}}{(\boldsymbol{\beta^*}^\top \Sigma \boldsymbol{\beta^*})^{\frac{1}{2}}} \left( \frac{\boldsymbol{\xi}^\top \boldsymbol{\tau}}{(\boldsymbol{\xi}^\top \Sigma \boldsymbol{\xi})^{\frac{1}{2}}} - \frac{\boldsymbol{\beta^*}^\top \boldsymbol{\tau}}{(\boldsymbol{\beta^*}^\top \Sigma \boldsymbol{\beta^*})^{\frac{1}{2}}} \frac{\boldsymbol{\xi}^\top \Sigma \boldsymbol{\beta^*}}{(\boldsymbol{\beta^*}^\top \Sigma \boldsymbol{\beta^*})^{\frac{1}{2}} (\boldsymbol{\xi}^\top \Sigma \boldsymbol{\xi})^{\frac{1}{2}}} \right)$$

$$> \frac{(\boldsymbol{\xi}^\top \Sigma \boldsymbol{\xi})^{\frac{1}{2}}}{(\boldsymbol{\beta^*}^\top \Sigma \boldsymbol{\beta^*})^{\frac{1}{2}}} \left( \frac{\boldsymbol{\beta^*}^\top \boldsymbol{\tau}}{(\boldsymbol{\beta^*}^\top \Sigma \boldsymbol{\beta^*})^{\frac{1}{2}}} - \frac{\boldsymbol{\beta^*}^\top \boldsymbol{\tau}}{(\boldsymbol{\beta^*}^\top \Sigma \boldsymbol{\beta^*})^{\frac{1}{2}}} \frac{\boldsymbol{\xi}^\top \Sigma \boldsymbol{\beta^*}}{(\boldsymbol{\beta^*}^\top \Sigma \boldsymbol{\beta^*})^{\frac{1}{2}} (\boldsymbol{\xi}^\top \Sigma \boldsymbol{\xi})^{\frac{1}{2}}} \right)$$

$$= \frac{(\boldsymbol{\xi}^\top \Sigma \boldsymbol{\xi})^{\frac{1}{2}}}{(\boldsymbol{\beta^*}^\top \Sigma \boldsymbol{\beta^*})^{\frac{1}{2}}} \frac{\boldsymbol{\beta^*}^\top \boldsymbol{\tau}}{(\boldsymbol{\beta^*}^\top \Sigma \boldsymbol{\beta^*})^{\frac{1}{2}}} \left( 1 - \frac{\boldsymbol{\xi}^\top \Sigma \boldsymbol{\beta^*}}{(\boldsymbol{\beta^*}^\top \Sigma \boldsymbol{\beta^*})^{\frac{1}{2}} (\boldsymbol{\xi}^\top \Sigma \boldsymbol{\xi})^{\frac{1}{2}}} \right)$$

$$\geq 0,$$

where we used the assumption $\frac{\boldsymbol{\xi}^\top \boldsymbol{\tau}}{(\boldsymbol{\xi}^\top \Sigma \boldsymbol{\xi})^{\frac{1}{2}}} > \frac{\boldsymbol{\beta^*}^\top \boldsymbol{\tau}}{(\boldsymbol{\beta^*}^\top \Sigma \boldsymbol{\beta^*})^{\frac{1}{2}}}$ for the first inequality and $\boldsymbol{\beta^*}^\top \boldsymbol{\tau} \geq 0$ and the Cauchy-Schwarz inequality to arrive at the last line. Since, by assumption, $\xi_u \geq 0$ for all $u$, this implies $\left. \frac{\partial}{\partial \beta_u} \frac{\boldsymbol{\beta}^\top \boldsymbol{\tau}}{(\boldsymbol{\beta}^\top \Sigma \boldsymbol{\beta})^{\frac{1}{2}}} \right|_{\boldsymbol{\beta}=\boldsymbol{\beta^*}} > 0$ for some $u$ and thus is a contradiction to Condition 1.

ii) Assume $\boldsymbol{\beta^*}^\top \boldsymbol{\tau} < 0$. We define $u^* = \underset{u \in [d]}{\mathrm{argmax}}\, \frac{\tau_u}{(e_u^\top \Sigma e_u)^{\frac{1}{2}}}$. By the third condition and the assumption $\boldsymbol{\beta^*}^\top \boldsymbol{\tau} < 0$, we have $0 > \frac{\boldsymbol{\beta^*}^\top \boldsymbol{\tau}}{(\boldsymbol{\beta^*}^\top \Sigma \boldsymbol{\beta^*})^{\frac{1}{2}}} \geq \frac{\tau_{u^*}}{\left(e_{u^*}^\top \Sigma e_{u^*}\right)^{\frac{1}{2}}}$. This implies $\tau_{u^*} < 0$. We then get

$$\frac{\boldsymbol{\xi}^\top \boldsymbol{\tau}}{(\boldsymbol{\xi}^\top \Sigma \boldsymbol{\xi})^{\frac{1}{2}}} = \sum_{u \in [d]} \xi_u \frac{\tau_u}{(\boldsymbol{\xi}^\top \Sigma \boldsymbol{\xi})^{\frac{1}{2}}} = \sum_{u \in [d]} \xi_u \frac{\tau_u \left(e_u^\top \Sigma e_u\right)^{\frac{1}{2}}}{(\boldsymbol{\xi}^\top \Sigma \boldsymbol{\xi})^{\frac{1}{2}} \left(e_u^\top \Sigma e_u\right)^{\frac{1}{2}}}$$

$$\leq \sum_{u \in [d]} \xi_u \frac{\tau_{u^*} \left(e_u^\top \Sigma e_u\right)^{\frac{1}{2}}}{\left(e_{u^*}^\top \Sigma e_{u^*}\right)^{\frac{1}{2}} (\boldsymbol{\xi}^\top \Sigma \boldsymbol{\xi})^{\frac{1}{2}}}$$

$$= \frac{\tau_{u^*}}{\left(e_{u^*}^\top \Sigma e_{u^*}\right)^{\frac{1}{2}}} \frac{\sum_{u \in [d]} \xi_u \left(e_u^\top \Sigma e_u\right)^{\frac{1}{2}}}{(\boldsymbol{\xi}^\top \Sigma \boldsymbol{\xi})^{\frac{1}{2}}}$$

$$\leq \frac{\tau_{u^*}}{\left(e_{u^*}^\top \Sigma e_{u^*}\right)^{\frac{1}{2}}} \leq \frac{\boldsymbol{\beta^*}^\top \boldsymbol{\tau}}{(\boldsymbol{\beta^*}^\top \Sigma \boldsymbol{\beta^*})^{\frac{1}{2}}},$$

where to arrive at the last line we used $\tau_{u^*} < 0$ and the triangle inequality $\sum_{u \in [d]} \xi_u \left(e_u^\top \Sigma e_u\right)^{\frac{1}{2}} = \sum_{u \in [d]} \xi_u \|\Sigma^{\frac{1}{2}} e_u\| \geq \| \sum_{u \in [d]} \xi_u \Sigma^{\frac{1}{2}} e_u \| = \|\Sigma^{\frac{1}{2}} \boldsymbol{\xi}\| = (\boldsymbol{\xi}^\top \Sigma \boldsymbol{\xi})^{\frac{1}{2}}$. Thus this violates the assumption $\frac{\boldsymbol{\xi}^\top \boldsymbol{\tau}}{(\boldsymbol{\xi}^\top \Sigma \boldsymbol{\xi})^{\frac{1}{2}}} > \frac{\boldsymbol{\beta^*}^\top \boldsymbol{\tau}}{(\boldsymbol{\beta^*}^\top \Sigma \boldsymbol{\beta^*})^{\frac{1}{2}}}$.

Note that the above inequalities also hold for $\frac{\boldsymbol{\beta^*}^\top \boldsymbol{\tau}}{(\boldsymbol{\beta^*}^\top \Sigma \boldsymbol{\beta^*})^{\frac{1}{2}}}$. Thus we get that $\frac{\boldsymbol{\beta^*}^\top \boldsymbol{\tau}}{(\boldsymbol{\beta^*}^\top \Sigma \boldsymbol{\beta^*})^{\frac{1}{2}}} = \frac{\tau_{u^*}}{\left(e_{u^*}^\top \Sigma e_{u^*}\right)^{\frac{1}{2}}}$. This implies that $l = |\mathcal{U}| = 1$. Thus the following statements hold true:

$$i) \qquad \boldsymbol{\beta^*}^\top \boldsymbol{\tau} < 0 \quad \Rightarrow \quad l = 1, \tag{14}$$

$$ii) \qquad l \geq 2 \qquad \Rightarrow \quad \boldsymbol{\beta^*}^\top \boldsymbol{\tau} \geq 0. \tag{15}$$

$\square$

## A.2   Gradient of objective

We overload the notation and define $\boldsymbol{z} := \boldsymbol{\tau} - \Sigma\boldsymbol{\beta}\frac{\boldsymbol{\beta}^\top\boldsymbol{\tau}}{\boldsymbol{\beta}^\top\Sigma\boldsymbol{\beta}}$ similar as in (7) but for any $\boldsymbol{\beta}$. Then

$$
\begin{aligned}
\nabla_{\boldsymbol{\beta}} f(\boldsymbol{\beta}) &= \nabla_{\boldsymbol{\beta}} \left( \frac{\boldsymbol{\beta}^\top\boldsymbol{\tau}}{(\boldsymbol{\beta}^\top\Sigma\boldsymbol{\beta})^{\frac{1}{2}}} \right) \\
&= \frac{(\boldsymbol{\beta}^\top\Sigma\boldsymbol{\beta})^{\frac{1}{2}}\nabla_{\boldsymbol{\beta}}(\boldsymbol{\beta}^\top\boldsymbol{\tau}) - \boldsymbol{\beta}^\top\boldsymbol{\tau}\nabla_{\boldsymbol{\beta}}((\boldsymbol{\beta}^\top\Sigma\boldsymbol{\beta})^{\frac{1}{2}})}{\boldsymbol{\beta}^\top\Sigma\boldsymbol{\beta}} \\
&= \frac{(\boldsymbol{\beta}^\top\Sigma\boldsymbol{\beta})^{\frac{1}{2}}\boldsymbol{\tau} - \frac{1}{2}\boldsymbol{\beta}^\top\boldsymbol{\tau}((\boldsymbol{\beta}^\top\Sigma\boldsymbol{\beta})^{-\frac{1}{2}}) \cdot 2\boldsymbol{\beta}^\top\Sigma}{\boldsymbol{\beta}^\top\Sigma\boldsymbol{\beta}} \\
&= \frac{1}{(\boldsymbol{\beta}^\top\Sigma\boldsymbol{\beta})^{\frac{1}{2}}} \left( \boldsymbol{\tau} - \Sigma\boldsymbol{\beta}\left( \frac{\boldsymbol{\beta}^\top\boldsymbol{\tau}}{(\boldsymbol{\beta}^\top\Sigma\boldsymbol{\beta})} \right) \right) \\
&= \frac{1}{(\boldsymbol{\beta}^\top\Sigma\boldsymbol{\beta})^{\frac{1}{2}}}\boldsymbol{z}.
\end{aligned}
\tag{16}
$$

## A.3   Proof of Equation (11)

In the proof of Theorem 1 we used that $\tilde{X}$ and $\boldsymbol{z}$ are independent. Which we prove here. Since $\tilde{X}$ and $\boldsymbol{z}$ are jointly normal, we only need to show that they are uncorrelated. To do so recall that we are only interested in the distribution under the null and hence $\boldsymbol{0} = \mathbb{E}\left[\boldsymbol{\tau}\right] = \mathbb{E}\left[\tilde{X}\right] = \mathbb{E}\left[\boldsymbol{z}\right]$. Since $\tilde{X}_u = 0$ for all $u \notin \mathcal{U}$ and $z'_u = 0$ for all $u' \in \mathcal{U}$, it suffices to show that $\tilde{X}_j$ is uncorrelated with $z_i$ for all $i \notin \mathcal{U}, j \in \mathcal{U}$.

$$
\begin{aligned}
\mathrm{Cov}\left[z_i, \tilde{X}_j\right] &= \mathbb{E}\left[z_i\tilde{X}_j\right] = \mathbb{E}\left[ \left(\tau_i - (\Sigma\bar{\Sigma}\boldsymbol{\tau})_i\right) ((\bar{\Sigma})^{\frac{1}{2}}\boldsymbol{\tau})_j \right] \\
&= \sum_{u\in\mathcal{U}}((\bar{\Sigma})^{\frac{1}{2}})_{ju}\mathbb{E}\left[\tau_i, \tau_u\right] - \sum_{s,t,u\in\mathcal{U}}((\bar{\Sigma})^{\frac{1}{2}})_{ju}\Sigma_{is}\bar{\Sigma}_{st}\mathbb{E}\left[\tau_t\tau_u\right] \\
&= \sum_{u\in\mathcal{U}}((\bar{\Sigma})^{\frac{1}{2}})_{ju}\Sigma_{iu} - \sum_{s,t,u\in\mathcal{U}}((\bar{\Sigma})^{\frac{1}{2}})_{ju}\Sigma_{is}\bar{\Sigma}_{st}\Sigma_{tu} \\
&= \left((\bar{\Sigma})^{\frac{1}{2}}\Sigma\right)_{ji} - \left(\Sigma\bar{\Sigma}\Sigma(\bar{\Sigma})^{\frac{1}{2}}\right)_{ij} \\
&= \left((\bar{\Sigma})^{\frac{1}{2}}\Sigma\right)_{ji} - \left(\Sigma(\bar{\Sigma})^{\frac{1}{2}}\right)_{ij} = 0.
\end{aligned}
$$

Thus $\tilde{X}$ and $\boldsymbol{z}$ are uncorrelated and independent.

# B   Solution of the continuous optimization problem

The presented solution is similarly described in Gretton et al. [4, Sec. 4]. There an $L1$ norm constraint was used, which, however does not change anything. For completeness we include it here. We define

$$
f(\boldsymbol{\beta}) := \frac{\boldsymbol{\beta}^\top\boldsymbol{\tau}}{(\boldsymbol{\beta}^\top\Sigma\boldsymbol{\beta})^{\frac{1}{2}}},
$$

and we want to find

$$
\boldsymbol{\beta}^* = \underset{\boldsymbol{\beta}\geq\boldsymbol{0}, \|\boldsymbol{\beta}\|=1}{\mathrm{argmax}} \frac{\boldsymbol{\beta}^\top\boldsymbol{\tau}}{(\boldsymbol{\beta}^\top\Sigma\boldsymbol{\beta})^{\frac{1}{2}}}.
$$

Since $f$ is a homogeneous function of order 0 in $\boldsymbol{\beta}$ we have $f(c\boldsymbol{\beta}) = f(\boldsymbol{\beta})$ for any $c > 0$. We can thus solve the relaxed problem (we implicitly exclude $\boldsymbol{\beta} = \boldsymbol{0}$)

$$
\boldsymbol{\beta}' = \underset{\boldsymbol{\beta}\geq\boldsymbol{0}}{\mathrm{argmax}} f(\boldsymbol{\beta}).
$$

The solution of the original problem is then simply given as a rescaled version of the relaxed problem $\boldsymbol{\beta}^* = \frac{\boldsymbol{\beta}'}{\|\boldsymbol{\beta}'\|}$. We shall solve the relaxed problem for two different cases.

  i) $\exists u \in [d] : \tau_u \geq 0$.

   In this case, we know that $\max_{\boldsymbol{\beta} \geq \mathbf{0}} f(\boldsymbol{\beta}) \geq 0$ and hence $\boldsymbol{\beta}' = \operatorname*{argmax}_{\boldsymbol{\beta} \geq \mathbf{0}} f(\boldsymbol{\beta}) \Leftrightarrow \boldsymbol{\beta}' = \operatorname*{argmax}_{\substack{\boldsymbol{\beta} \geq \mathbf{0} \\ f(\boldsymbol{\beta}) \geq 0}} f^2(\boldsymbol{\beta})$. The set $S := \{\boldsymbol{\beta} \in \mathbb{R}^d | \boldsymbol{\beta} \geq \mathbf{0}, f(\boldsymbol{\beta}) \geq 0\}$ is convex and the functions $g_1(\boldsymbol{\beta}) := (\boldsymbol{\beta}^\top \boldsymbol{\tau})^2$ and $g_2(\boldsymbol{\beta}) := \boldsymbol{\beta}^\top \Sigma \boldsymbol{\beta}$ are convex (recall that $\Sigma$ is a positive matrix). Thus our problem becomes

$$\boldsymbol{\beta}' = \operatorname*{argmax}_{\boldsymbol{\beta} \in S} \frac{g_1(\boldsymbol{\beta})}{g_2(\boldsymbol{\beta})},$$

   which is a concave fractional program. In our implementation we solve it by fixing $\boldsymbol{\beta}^\top \boldsymbol{\tau} = a$ for some $a > 0$ and then minimizing the denominator. Thus we are solving the quadratic optimization problem

$$\begin{aligned} \text{minimize} \quad & \boldsymbol{\beta}^\top \Sigma \boldsymbol{\beta} \\ \text{subject to:} \quad & \boldsymbol{\beta} \geq \mathbf{0} \\ & \boldsymbol{\beta}^\top \boldsymbol{\tau} = a. \end{aligned}$$

   We solve this problem with the CVXOPT python package [43].

  ii) $\tau_u < 0 \, \forall u \in [d]$.

   In this case we have $\boldsymbol{\beta}^{*\top} \boldsymbol{\tau} < 0$. By (15) we have $l = 1$. Thus we simply $\boldsymbol{\beta}^* = e_{u^*}$, where $u^* = \operatorname*{argmax}_{u \in [d]} \frac{\tau_u}{\Sigma_{u,u}}$.

Note that in the case $\boldsymbol{\tau} = \mathbf{0}$, $\boldsymbol{\beta}^*$ is not well defined and we could randomly select any $\boldsymbol{\beta}^*$. However, the probability of this happening is 0.

## C  Other proofs

### C.1  Proof of Corollary 1

As we pointed out in the main paper, when selecting a test from a countable number of test that can be written as projections of the base tests $\boldsymbol{\tau}$ we can use the results of Lee et al. [24]. For completeness we explicitly include the relevant theorem.

**Theorem 2** (Polyhedral Lemma [24], Theorem 5.2). *Let $\boldsymbol{\tau} \sim \mathcal{N}(\boldsymbol{\mu}, \Sigma)$, $\boldsymbol{\eta}, \boldsymbol{\mu} \in \mathbb{R}^d$, $\Sigma \in \mathbb{R}^{d \times d}$ positive definite, and $A \in \mathbb{R}^{s \times d}$, $\boldsymbol{b} \in \mathbb{R}^s$ for some $s \in \mathbb{N}$. Define $\boldsymbol{c} := \Sigma \boldsymbol{\eta} \left( \boldsymbol{\eta}^\top \Sigma \boldsymbol{\eta} \right)^{-1}$ and $\boldsymbol{z} := \left( I_d - \boldsymbol{c} \boldsymbol{\eta}^\top \right) \boldsymbol{\tau}$. Then we have*

$$\left[ \boldsymbol{\eta}^\top \boldsymbol{\tau} | A \boldsymbol{\tau} \leq \boldsymbol{b}, \boldsymbol{z} = \hat{\boldsymbol{z}} \right] \stackrel{d}{=} TN \left( \boldsymbol{\eta}^\top \boldsymbol{\mu}, \boldsymbol{\eta}^\top \Sigma \boldsymbol{\eta}, \mathcal{V}^-(\hat{\boldsymbol{z}}), \mathcal{V}^+(\hat{\boldsymbol{z}}) \right),$$

*where $TN(\mu, \sigma^2, a, b)$ denotes a Gaussian distribution with mean $\mu$ and variance $\sigma^2$ that is truncated at $a$ and $b$. Here*

$$\mathcal{V}^-(\boldsymbol{z}) := \max_{j:(A\boldsymbol{c})_j < 0} \frac{\boldsymbol{b}_j - (A\boldsymbol{z})_j}{(A\boldsymbol{c})_j}, \quad \mathcal{V}^+(\boldsymbol{z}) := \min_{j:(A\boldsymbol{c})_j > 0} \frac{\boldsymbol{b}_j - (A\boldsymbol{z})_j}{(A\boldsymbol{c})_j}.$$

Note that $\boldsymbol{c}$ is simply a fixed vector. $\boldsymbol{z}$ is a random variable that can be shown to be independent of $\boldsymbol{\eta}^\top \boldsymbol{\tau}$. The result enables us to draw a realization $\hat{\boldsymbol{\tau}}$ of the random variable (RV) $\boldsymbol{\tau}$ and select $\boldsymbol{\eta}$ if $A\hat{\boldsymbol{\tau}} \leq \boldsymbol{b}$. Since the truncation points of the Gaussian only depend on $\hat{\boldsymbol{z}}$, and $\boldsymbol{z}$ is independent of $\boldsymbol{\eta}^\top \boldsymbol{\tau}$, we can compute a reliable $p$-value of $\boldsymbol{\eta}^\top \hat{\boldsymbol{\tau}}$ by using Theorem (2).

*Proof of Corollary 1.* We need the distribution of $\frac{\tau_{u^*}}{\sigma_{u^*}}$ after conditioning on the selection of $u^*$. To obtain this distribution we first need to characterize the event that leads to the selection of $u^*$. The

selection event simply is $u^* = \underset{u \in [d]}{\operatorname{argmax}} \frac{\tau_u}{\sigma_u} \Leftrightarrow \frac{\tau_{u^*}}{\sigma_{u^*}} \geq \frac{\tau_u}{\sigma_u}$ for all $u \in [d]$. Therefore, define the matrix
$A := \operatorname{diag}(\frac{1}{\sigma_1}, \ldots, \frac{1}{\sigma_d}) - \frac{1}{\sigma_{u^*}} A(u^*)$, where $\operatorname{diag}(\cdot)$ defines a $d \times d$ matrix with the arguments on its diagonal and zeros everywhere else and $A(\cdot)$ is a $d \times d$ matrix with ones in the column given by its argument and zeros everywhere else. It follows that $(A\boldsymbol{\tau})_j = \frac{\tau_j}{\sigma_j} - \frac{\tau_{u^*}}{\sigma_{u^*}}$, and $u^* = \underset{u \in [d]}{\operatorname{argmax}} \frac{\tau_u}{\sigma_u}$ is equivalent to $A\boldsymbol{\tau} \leq \mathbf{0} =: \boldsymbol{b}$. Apart from this we define $\boldsymbol{\eta} := \frac{e_{u^*}}{\sigma_{u^*}}$, so that $\boldsymbol{\eta}^\top \boldsymbol{\tau} = \frac{\tau_{u^*}}{\sigma_{u^*}}$. Then we can define $\boldsymbol{c} := \Sigma \boldsymbol{\eta} \left( \boldsymbol{\eta}^\top \Sigma \boldsymbol{\eta} \right)^{-1}$ and $\boldsymbol{z} := \left( I_d - \boldsymbol{c}\boldsymbol{\eta}^\top \right) \boldsymbol{\tau}$ as in Theorem 2, and denote by $\hat{\boldsymbol{z}}$ the value of the random variable $\boldsymbol{z}$ that we observed (note that this coincides with the definition we used for $\boldsymbol{z}$ in the Corollary). By our definitions we have $(A\boldsymbol{c})_j = \frac{\Sigma_{ju^*}/\sigma_j - \sigma_{u^*}}{\sigma_{u^*}} = \frac{1}{\sigma_{u^*}\sigma_j} (\Sigma_{u^*j} - \sigma_u^* \sigma_j)$. Since $\Sigma$ is positive definite, $(A\boldsymbol{c})_j < 0$ if $j \neq u^*$ and $(A\boldsymbol{c})_{u^*} = 0$. Thus according to Theorem 2, $\mathcal{V}^+$ is an optimization over an empty set and we can set it to $\infty$. Further $(A\boldsymbol{z})_j = \frac{1}{\sigma_{u^*}\sigma_j} \left( \tau_j \sigma_{u^*} - \frac{\Sigma_{u^*j}}{\sigma_{u^*}} \tau_{u^*} \right)$.

Combining the previous two expressions we obtain $\frac{-(A\boldsymbol{z})_j}{(A\boldsymbol{c})_j} = \frac{\tau_j \sigma_{u^*} - \frac{\Sigma_{u^*j}}{\sigma_{u^*}} \tau_{u^*}}{\sigma_u^* \sigma_j - \Sigma_{u^*j}} = \frac{\sigma_{u^*} z_j}{\sigma_u^* \sigma_j - \Sigma_{u^*j}}$. We can then directly apply Theorem 2 and the result follows. □

## C.2 Proof of Equation (3)

In the main paper we omitted the proof of the closed form solution of $\boldsymbol{\beta}^\infty$. We thus need to show

$$\underset{\|\boldsymbol{\beta}\|=1}{\operatorname{argmax}} \frac{\boldsymbol{\beta}^\top \boldsymbol{\mu}}{(\boldsymbol{\beta}^\top \Sigma \boldsymbol{\beta})^{\frac{1}{2}}} = \frac{\Sigma^{-1}\boldsymbol{\mu}}{\|\Sigma^{-1}\boldsymbol{\mu}\|}.$$

*Proof.* We are only interested in $\boldsymbol{\beta}^\infty$ if the alternative hypothesis is true and thus at least one entry of $\boldsymbol{\mu}$ is positive. We further assume that the covariance $\Sigma$ has full rank. Hence there exists a $b > 0$ such that $\boldsymbol{\beta}^\top \Sigma \boldsymbol{\beta} > b$ for all $\boldsymbol{\beta}$ with $\|\boldsymbol{\beta}\| = 1$, i.e., the denominator $(\boldsymbol{\beta}^\top \Sigma \boldsymbol{\beta})^{\frac{1}{2}}$ is strictly positive and has a lower bound. Since $\boldsymbol{\mu} \neq \mathbf{0}$, this implies that $\max_{\|\boldsymbol{\beta}\|=1} \frac{\boldsymbol{\beta}^\top \boldsymbol{\mu}}{(\boldsymbol{\beta}^\top \Sigma \boldsymbol{\beta})^{\frac{1}{2}}} > 0$. Also the nominator has an upper bound which is given by $\boldsymbol{\beta}^\top \boldsymbol{\mu} \leq \boldsymbol{\mu}^\top \boldsymbol{\mu} / \|\boldsymbol{\mu}\|$ if $\|\boldsymbol{\beta}\| = 1$. Hence the whole maximization is upper bounded. Since the unit sphere in $\mathbb{R}^d$ is a compact set, we can conclude that the maximum of the objective is attained. Thus it suffices to show that for all $\boldsymbol{\beta} \neq \boldsymbol{\beta}^\infty$ the objective is not maximized. In the following, we use that the objective of the maximization is a homogeneous function of order 0 in $\boldsymbol{\beta}$ and hence we can relax the constraint $\|\boldsymbol{\beta}\| = 1$ to $\boldsymbol{\beta} \neq \mathbf{0}$ (note that this not affect the existence of the maximum). As we showed in Appendix A.2, the gradient of the objective function is given by

$$\nabla_{\boldsymbol{\beta}} \frac{\boldsymbol{\beta}^\top \boldsymbol{\mu}}{(\boldsymbol{\beta}^\top \Sigma \boldsymbol{\beta})^{\frac{1}{2}}} = \frac{1}{(\boldsymbol{\beta}^\top \Sigma \boldsymbol{\beta})^{\frac{1}{2}}} \left( \boldsymbol{\mu} - \Sigma \boldsymbol{\beta} \left( \frac{\boldsymbol{\beta}^\top \boldsymbol{\mu}}{(\boldsymbol{\beta}^\top \Sigma \boldsymbol{\beta})} \right) \right).$$

Setting the gradient to zero we obtain

$$\nabla_{\boldsymbol{\beta}} \frac{\boldsymbol{\beta}^\top \boldsymbol{\mu}}{(\boldsymbol{\beta}^\top \Sigma \boldsymbol{\beta})^{\frac{1}{2}}} = \mathbf{0} \Leftrightarrow \boldsymbol{\beta} = c \cdot \Sigma^{-1}\boldsymbol{\mu} \text{ for some } c \in \mathbb{R}.$$

If $c < 0$ the objective attains a negative value, since $\Sigma^{-1}$ is a strictly positive matrix, and thus does not correspond to the global maximum, which we already know to be positive. Thus, the maximum has to be attained for some $c > 0$. Using the constraint $\|\boldsymbol{\beta}\| = 1$ it follows that the global optimum is attained at $\boldsymbol{\beta}^\infty$. □

# D Experimental details and further experiments

We first give some details on the experiments we showed in the main paper. For all the experiments we start with a set of $d$ base kernels $\mathcal{K} = [k_1, \ldots, k_d]$ that are chosen independently of the observed data samples $X = \{x_1, \ldots, x_{2n}\} \sim P^{2n}$ and $Y = \{y_1, \ldots, y_{2n}\} \sim Q^{2n}$. First, we define $z_i := (x_i, x_{n+i}, y_i, y_{n+i})$ and compile $X$ and $Y$ into $Z = \{z_1, \ldots, z_n\}$. For each kernel we define $h_i(z) := h_i(x, x', y, y') := k_i(x, x') + k_i(y, y') - k_i(x, y') - k_i(y, x')$. For all the methods we estimate the covariance matrix on the whole dataset as

$$\hat{\Sigma}_{ij} = \frac{1}{n} \sum_{k=1}^{n} h_i(z_k) h_j(z_k) - \frac{1}{n} \sum_{k=1}^{n} h_i(z_k) \frac{1}{n} \sum_{k'=1}^{n} h_j(z_{k'}).$$

We then further assume that $\Sigma = \hat{\Sigma}$ which is justified since the CLT also works with a consistent estimate of the covariance. For all the methods that do not split the data (OST, WALD, and NAIVE) we estimate the entries of $\hat{\boldsymbol{\tau}}$ as

$$\hat{\tau}_i = \sqrt{n}\, \widehat{\mathrm{MMD}}_{\mathrm{lin}}^2(P, Q) = \sqrt{n}\, \frac{1}{n} \sum_{k=1}^{n} h_i(z_k),$$

i.e., we directly absorb the $\sqrt{n}$ dependence of the asymptotic distribution into $\boldsymbol{\tau}$. For data splitting we estimate $\hat{\boldsymbol{\tau}}_{\mathrm{tr}}$ on a split of the data and $\hat{\boldsymbol{\tau}}_{\mathrm{te}}$ on the other split. For example SPLIT0.3 means that $30\%$ of the data are used to estimate $\hat{\boldsymbol{\tau}}_{\mathrm{tr}}$ and $70\%$ used to estimate $\hat{\boldsymbol{\tau}}_{\mathrm{te}}$. We assume that the number of samples in the respective subsets are even and otherwise neglect some samples.

**Methods**   We compare four different methods:

i) OST: The test we recommend to use, as described in Algorithm 1.

ii) WALD: The Wald test, which does not take into account the prior information $\boldsymbol{\mu} \geq \mathbf{0}$.

iii) SPLIT: Data splitting similar to the approach in Gretton et al. [4]. SPLIT0.3 denotes that $30\%$ of the data are used for learning $\boldsymbol{\beta}^*$ and $70\%$ are used for testing. Here we first, learn $\boldsymbol{\beta}^*$ on the training sample, i.e., $\boldsymbol{\beta}^* = \underset{\|\Sigma\boldsymbol{\beta}\|=1, \Sigma\boldsymbol{\beta} \geq 0}{\operatorname{argmax}} \frac{\boldsymbol{\beta}^\top \boldsymbol{\tau}_{tr}}{(\boldsymbol{\beta}^\top \Sigma\boldsymbol{\beta})^{\frac{1}{2}}}$. We then use the test statistic $\frac{\hat{\boldsymbol{\beta}}^\top \boldsymbol{\tau}_{te}}{(\hat{\boldsymbol{\beta}}^\top \Sigma\hat{\boldsymbol{\beta}})^{\frac{1}{2}}}$, which follows a standard normal under the null. This differs from the approach in Gretton et al. [4], since we optimize with the constraints $\Sigma\boldsymbol{\beta} \geq \mathbf{0}$, whereas Gretton et al. [4] suggested a simple positivity constraint $\boldsymbol{\beta} \geq \mathbf{0}$. We discuss this in Section D.2.

iv) NAIVE: Two stage procedure where all the data is used for learning and testing without correcting for the dependency, i.e., without splitting the data. Thus the test statistic is the same as for OST, but we work with the wrong null distribution, i.e., the one that is only valid for data splitting. This approach is not a well-calibrated test, see Fig. 8 and hence is useless.

**Datasets**   The `DIFF VAR` dataset is a simple one-dimensional toy dataset, where $P = \mathcal{N}(0, 1)$ and $Q = \mathcal{N}(0, 1.5)$.

The `Blobs` dataset was constructed using a mixture of 2D Gaussians on a $3 \times 3$ grid. The centers of the Gaussians are set to $\mu_1, \ldots, \mu_9 = (0, 0), (0, 1), (0, 2), (1, 0), (1, 1), (1, 2), (2, 0), (2, 1), (2, 2)$ and the covariances are $\Sigma_P = \mathrm{diag}(0.1, 0.3)$ and $\Sigma_Q = \mathrm{diag}(0.3, 0.1)$. Samples from $P$ and $Q$ are shown in Figure 5. The `Blobs` dataset is constructed such that the main variance in the data does not reflect the difference between $P$ and $Q$, which happens on a smaller length scale. This is inspired by Gretton et al. [4], where similar data has been considered to showcase that such problems benefit from careful kernel choice. We can reproduce this behavior with our results, which show that for this dataset the performance is bad if one only considers the median heuristic Gaussian kernel together with a linear kernel.

The `MNIST` dataset was constructed by first downsampling all the images to $7 \times 7$ pixels (originally $28 \times 28$), by simply averaging over fields of $4 \times 4$ pixels. We define $P$ to contain all the digits, while $Q$ only contains uneven digits. For our experiments we draw with replacement from the images in the database. Some samples from both distributions are shown in Figure 6.

**Experiments for Figure 3**   For Figure 3 we constructed a 1-D data set such that both $P$ and $Q$ are symmetric (thus all uneven moments vanish) and have the same variance, see Figure 7.

## D.1   Type-I errors

To verify which methods are theoretically justified, i.e., control the Type-I error at a level $\alpha = 0.05$, we run the following experiments, similar to the experiments in the main paper, where $P = Q$.

1. DIFF VAR ($p = 1$): $P = \mathcal{N}(0, 1)$ and $Q = \mathcal{N}(0, 1)$.

2. MNIST ($p = 49$): We consider downsampled 7x7 images of the MNIST dataset [40], where $P$ contains all the digits and $Q = P$.

Figure 5: Samples from BLOBS dataset.

Figure 6: Samples from downsampled MNIST dataset. $P$ (left) contains all digits, while $Q$ (right) only contains uneven digits.

3. BLOBS ($p = 2$): A mixture of anisotropic Gaussians and $P = Q$.

The results are in Figure 8. All the methods except NAIVE correctly control the Type-I error at a rate $\alpha = 0.05$ even for relatively small sample sizes. Note that all the described approaches rely on the asymptotic distribution. The critical sample size, at which it is safe to use, generally depends on the distributions $P$ and $Q$ and also the kernel functions. A good approach to simulating Type-I errors in in two-sample testing problems is to merge the samples and then randomly split them again. If the estimated Type-I error is significantly larger that $\alpha$, working with the asymptotic distribution is not reliable.

## D.2 Comparison of the constraints

In Section 3.2 we motivate to constrain the set of considered $\beta$ to obey $\Sigma\beta \geq 0$, thus incorporating the knowledge $\mu \geq 0$. All our experiments suggest that this constraint indeed improves test power as compared to the general Wald test. In Gretton et al. [4] a different constraint was chosen. There $\beta$ is constrained to be positive, i.e., $\beta \geq 0$. The motivation for their constraint is that the sum of positive definite (pd) kernel functions is again a pd kernel function [21]. Thus, by constraining $\beta \geq 0$ one ensures that $k = \sum_{u=1}^{d} \beta_u k_u$ is also a pd kernel. While this is sensible from a kernel perspective, it is unclear whether this is smart from a hypothesis testing viewpoint. From the latter perspective we do not necessarily care whether or not $\beta^*$ defines a pd kernel. Our approach instead was purely motivated to increase test power over the Wald test. In Figure 9 we thus compare the two different

Figure 7: Probability density functions used for the experiment in Figure 3 of the main paper. Both distributions are symmetric and are constructed to have the same variance.

Figure 8: Type-I errors for similar distributions as the one considered in the main paper. To simulate type-I errors we choose distributions $P = Q$ that are similar to the ones considered for the Type-II errors. We see that all well-calibrated methods reliably control the Type-I error at a rate $\alpha = 0.05$, and conclude that working with the asymptotic distributions is well justified for the considered examples. The NAIVE approach fails to control the error, as it overfits in the training phase without a correction in the testing phase.

constraints to the Wald test on the examples that were also investigated in the main paper with $d = 6$ kernels (again five Gaussian kernels and a linear kernel).

From Figure 9 we observe that the positivity constraint of Gretton et al. [4] does not allow for general conclusions. Depending on the problem, the positivity constraint can both lead to higher or lower test power than the Wald test or tests with the constraint $\Sigma\boldsymbol{\beta} \geq \mathbf{0}$. It will thus generally depend on the problem at hand which constraint is better. However, at least the approach we recommend ($\Sigma\boldsymbol{\beta} \geq \mathbf{0}$) seems to guarantee a test power at least as high as the Wald test, whereas the positivity constraint can also be worse. As long as one has not a clear indication that the positivity constraint leads to better performance, we thus recommend the constraint $\Sigma\boldsymbol{\beta} \geq \mathbf{0}$.

### D.3 Discrete selection from $T_{\mathbf{base}}$

In this experiment, we use the same datasets and base kernels as for the experiment in the main paper. Instead of considering $T_{\mathrm{Wald}}$ and $T_{\mathrm{OST}}$, we consider $T_{\mathrm{base}}$. We thus only compare to a data-splitting approach where also one of the base test statistics is selected. For completeness, we also include the NAIVE approach, which again overfits for $d > 1$. Note that the thresholds for $\tau_{\mathrm{base}}$ can be computed with Corollary 1 and do not rely on Theorem 1. The results are shown in Figure 10, again averaged over 5000 independent trials. In most of the cases, we observe that $\tau_{\mathrm{base}}$ outperforms the data-splitting

Figure 9: Comparison of the different constraints: In the main paper we argue that OST is a principled approach to constraint the class of considered tests, when $\boldsymbol{\mu} \geq \mathbf{0}$ is guaranteed. Gretton et al. [4] suggested a different constraint $\boldsymbol{\beta} \geq \mathbf{0}$. With Theorem 1, we can also work with these constraints without data-splitting. The results suggest that indeed OST is a meaningful way to constrain the class of tests, as it consistently outperforms the Wald test. On the other hand the constraint suggested by Gretton et al. [4], can only be seen as a heuristic. For some cases it performs better than the Wald test and the OST, but it can also perform worse.

approaches. However, for the MNIST dataset and $d = 2$, the splitting approach that uses $10\%$ for learning and $90\%$ for testing does perform slightly better. Our attempt to explain this behavior lies in the truncation $\mathcal{V}^-$ of the conditional distribution. While for OST, we can show that $\mathcal{V}^- \leq 0$ (see proof of Theorem 1), for Corollary 1, $\mathcal{V}^-$ cannot be bounded. If $\mathcal{V}^-$ is very large, the selected test is very conservative. We acknowledge that this is not a sufficient analysis of this phenomenon, but leave a more theoretical treatment for future work.

## E    Singular covariance matrices

In the main paper we assumed that $\Sigma$ is strictly positive, i.e., non-singular. However, in practice, some eigenvalues of the covariance matrix can be sufficiently close to zero to cause numerical problems. In the case of the kernel two-sample test, this can happen if we consider kernels that are too similar and thus cause redundancy in our observations. In practice, this happens for example if we consider Gaussian kernels with too similar bandwidths on an easy problem.

**Note on regularization:** One strategy to recover the numerical stability of the algorithm is to regularize the covariance matrix $\Sigma \to \Sigma + \lambda I$. Doing this indeed increases the numerical stability, since it leads to a well-behaved condition number. However, it also makes the whole approach more conservative, since the (artificially) increased variance decreases the value of the test statistic compared to the threshold. This leads to an increase of Type-II error and thus a loss of power. To evade this, we suggest the more elaborate strategy below.

Since $\Sigma$ is symmetric, there exists an orthonormal basis $\{v_i\}_{i \in [d]}$ and non-negative numbers $\{\lambda_i\}_{i \in [d]}$ such that

$$\Sigma = \sum_{i=1}^{d} \lambda_i v_i v_i^\top.$$

If $\Sigma$ is singular, we can assume WLOG that there exists $d_0 \in [d]$ such that $\lambda_i = 0$ if $i \leq d_0$ and hence

$$\Sigma = \sum_{i=d_0+1}^{d} \lambda_i v_i v_i^\top.$$

Now if $v_i^\top \boldsymbol{\tau} \neq 0$ for some $i \in [d_0]$, we immediately know that $\boldsymbol{\mu} \neq \mathbf{0}$ and could reject. In other words the signal-to-noise ratio along this direction is infinite. Thus, in the following we assume

Figure 10: Type-II errors for discrete selection, i.e., the class of considered tests is $T_{\text{base}}$. The rows (columns) correspond to different datasets (sets of base kernels). Similar as in Figure 2, our approach $\tau_{\text{base}}$ outperforms the splitting approaches in most cases. However, for the MNIST dataset and $d = 2$ we see that the splitting approach with $10\%$ training and $90\%$ testing data (SPLIT0.1) performs better.

$v_i^\top \boldsymbol{\tau} = 0$ for all $i \in [d_0]$, and hence, $\sum_{i=d_0+1}^{d} v_i v_i^\top \boldsymbol{\tau} = \boldsymbol{\tau}$. We can then rewrite the objective as follows

$$\max_{\Sigma\boldsymbol{\beta}\geq\mathbf{0}} \frac{\boldsymbol{\beta}^\top \boldsymbol{\tau}}{(\boldsymbol{\beta}^\top \Sigma \boldsymbol{\beta})^{\frac{1}{2}}} = \max_{\sum_{i=d_0+1}^{d} \lambda_i v_i v_i^\top \boldsymbol{\beta}\geq\mathbf{0}} \frac{\boldsymbol{\beta}^\top \sum_{i=d_0+1}^{d} v_i v_i^\top \boldsymbol{\tau}}{(\boldsymbol{\beta}^\top \sum_{i=d_0+1}^{d} \lambda_i v_i v_i^\top \boldsymbol{\beta})^{\frac{1}{2}}}.$$

Now define $\boldsymbol{\alpha} := \sum_{i=d_0+1}^{d} \lambda_i v_i v_i^\top \boldsymbol{\beta}$. Since $\Sigma$ is symmetric its pseudoinverse is given as $\Sigma^+ = \sum_{i=d_0+1}^{d} \frac{1}{\lambda_i} v_i v_i^\top$ and we get

$$\max_{\sum_{i=d_0+1}^{d} \lambda_i v_i v_i^\top \boldsymbol{\beta}\geq\mathbf{0}} \frac{\boldsymbol{\beta}^\top \sum_{i=d_0+1}^{d} v_i v_i^\top \boldsymbol{\tau}}{(\boldsymbol{\beta}^\top \sum_{i=d_0+1}^{d} \lambda_i v_i v_i^\top \boldsymbol{\beta})^{\frac{1}{2}}} = \max_{\boldsymbol{\alpha}\geq\mathbf{0}} \frac{\boldsymbol{\alpha}^\top \Sigma^+ \boldsymbol{\tau}}{(\boldsymbol{\beta}^\top \Sigma^+ \boldsymbol{\beta})^{\frac{1}{2}}}.$$

Similar as in Remark 1 we can define $\boldsymbol{\rho} := \Sigma^+ \boldsymbol{\tau}$ and $\Sigma' = \Sigma^+$. However, in Theorem 1 we assumed that the covariance is not singular. Therefore in Theorem 1 we used $l = |\mathcal{U}|$, which corresponded to the rank of $\Pi\Sigma\Pi$ (see Appendix A). However, in the present case the rank of $\Pi\Sigma^+\Pi$ does not equal the number of non-zero entries of $\boldsymbol{\beta}$. Therefore we use $l = \text{rank}(\Pi\Sigma^+\Pi)$. With this we can apply Theorem 1 and get the conditional distribution under the null.

In practice, we have to treat the covariance matrix as singular if its condition number is below some threshold, as otherwise the numerical precision does not suffice to invert matrices faithfully.