[Reviews · NeurIPS 2020]

Review 1

Summary and Contributions: Two sample testing is used to test whether two given sample sets come from the same distribution or not. Recent testing methods are parameterized giving a family of test statistic instead of just one. So, the question often becomes which is the best statistic to use. The best parameters are often chosen by splitting the data, tuning the parameters on one set and doing the actual test on the other. However, such splitting reduces the power of the test. This paper proposes an approach that does not require one to do the data splitting. The paper proposes an approach, wherein, it takes as input, the (pre-determined) family of statistics that one can calculate on the given data, then outputs the best test statistic. Thus, utilizing the whole dataset to do selection of parameter and testing together. In terms of contributions: 1] The paper proposes a novel approach to do statistical testing while utilizing the full dataset to tune parameters and test simultaneously. 2] The idea of considering a base set of statistics having joint normal distribution has been investigated before in the literature, this paper neatly connects that idea with the concept of data splitting. 3] Give theoretical analysis for the proposed method. While the discussion in the paper was restricted to kernel based methods of testing, the idea itself is much more broadly applicable. 4] Extends the PSI framework by Lee et. al. which was done for finite candidate sets to uncountable sets for the purposes of this problem.

Strengths: The theoretical analysis of the paper looks sound. The proof for the theorems and lemmas are given in the appendix and they looked right. Empirical evaluation looked good. I was able to run the code they provided on the datasets and reproduce the results. The results are very promising. The method is novel and the work is very significant given the recent interests in statistical hypothesis testing in several scenario, including its applicability to learning models like GAN. I believe that the work is very relevant to the NeurIPS community, where many of the prior works were already published in NeurIPS.

Weaknesses: The method requires one the determine the base set of statistics upfront. While this limitation has existed in prior works where one tune hyperparameter from a list of parameters. Newer methods like that Kirchler et. al. (cited in the paper) attempt to learn a representation for the point which can be used in MMD statistic, thus offering a newer way to learn kernels (apart from just selecting parameters from a list). It is unclear to me at least, whether the proposed approach can extend to these kinds of work. Two of the datasets utilized are toy datasets, which are great to establish the correctness of the test, but it would have been good to see couple of results on some other dataset. Also, it would have been good to check against some more statistical testing methods, particularly the follow up to MMD-u statistic work [A] which claimed lower sample complexity. [A] B-test: A Non-parametric, Low Variance Kernel Two-sample Test, Zaremba, W., Gretton, A., Blaschko, M. NIPS 2013

Correctness: The claims made in the paper look correct and the supporting proof looks sound under the assumptions made. Empirical methodology looks ok, albeit some more evaluations could have helped in establishing the value of the proposed approach.

Clarity: The paper is clear and understandable. It walks the readers from the basics to the proposed theorems and lemmas in a staged manner, making it a very educational read.

Relation to Prior Work: The paper discusses the connection with the prior work, particular the work done on post-selection inference. It also distinguishes itself from the prior work, not only in terms of how this idea is applied but also extends the results for PSI from finite candidate set to uncountable set. It also discusses the connection with the prior approaches to hypothesis testing based on tunable hyperparameters which relied on splitting data, and differentiates itself by proposing an approach that does not require any splitting.

Reproducibility: Yes

Additional Feedback: An update to the README to give some steps on how one can utilize their own data in the code would be very useful. --- Post Author Respose: I have read the author's response. One point raised in the review was that their approach requires pre-determined base set of statistics. With the world going towards deep learning based ideas for hypothesis testing, where does this work stand with respect to that? The authors have sufficiently responded to this question, by stating that in their current work, the selection event is well characterized which is not necessarily the case for deep learning based methods, at least not yet. The other point raised was about choosing a few more non-toy dataset for experimentation. Unfortunately, the authors haven't addressed this point at all in their response. Finally, a request was made to compare against B-test particularly given that B-test claims lower sample complexity and hence can derive higher test power from small data. However, the authors have claimed space constraint as a reason for not doing the comparison. But IMHO if one had to choose one method between linear MMD vs B-test to compare against, one might have chosen B-test. Regardless, I still feel positive about the work done in this paper and will retain my current evaluation.


Review 2

Summary and Contributions: In this paper, the authors adopt the normalized linear combinations of linear-time MMD as statistics to optimize the combination weights, which maximize the SNR and hence minimize the Type-II error. Three ways of combinations are considered including base, Wald and OST. In my opinion, the main contribution of this paper are the limiting distributions of the combined statistics, which are shown in Corollary 1 and Theorem 1. Using these limiting distributions, we can conduct the hypothesis test easily. However, I am confusing about the topic and motivation of this paper. The learning of the combination weights and the testing based on the obtained limiting distributions should be the cores of this paper. However, most discussions are concentrated on the problem of data splitting including the abstract and introduction. I think even when we split the data, your statistics can also be applied and the limiting distributions still work well on the training data. Am I right? --- After Authors' Feedback --- I have checked the feedback and the reviews of other reviewers. I decided to change my score to 7.

Strengths: The theoretical results of this paper are solid. The forms of the limiting distributions are attractive. They are just truncated normal distributions and chi square distributions. Using these kind of distributions, we can conduct the hypothesis testing much easier. These limiting distributions are also novel for the community.

Weaknesses: Empirical evaluations are generally enough. But I suggest compare more MMD-based statistics, even if the power of the combined statistics could be weaker than these statistics, since it is known to us that the combined statistics are based on linear-time MMD. Could the authors explain whether these kind of combinations can be extended to the GOF test, such as FSSD?

Correctness: The empirical methodology is correct. I didn't go much deeper into the proof of Theorem 1.

Clarity: The organization is very clear and the statements are well supported. The writing is generally good. But I am confusing about the topic and motivation.

Relation to Prior Work: The authors have clearly discussed the relations between this paper and previous works.

Reproducibility: Yes

Additional Feedback:


Review 3

Summary and Contributions: When practicing hypothesis testing, one often needs to select hyperparameters (e.g. kernel hyperparameters, coefficients of a linear combination between different test statistics). One common way to proceed is to split the data at disposal into 2 subsamples: one is used to select the adequate parameter, and the other to perform the test \it{per se} using the previously learned parameter. This method however suffers from the fact that the test is performed on a sample with significantly less observations than the original sample size, degrading the performance. The present work proposes a way to avoid splitting, and to learn the parameter and test on the whole sample each time, against additional efforts to deal with the dependency induced. Three scenarios of linear combinations among jointly normal base statistics are analyzed (base, Wald, ost), with explicit formulas given for the asymptotic distributions. Empirical experiments endorse the soundness of the approach.

Strengths: - the paper is clear and well written - the methodology is sound, and the motivation for not splitting the data is real and should benefit the community - I found the idea of leveraging the active set very interesting. It should maybe deserve better exposition through a detailed sketch of proof for instance - numerical experiments are conclusive

Weaknesses: - the paper is limited to the analysis of test statistics that write as linear combinations of base test statistics. This is in my opinion the main drawback of this work, but I might not be sufficiently aware of the literature to conclude if this point is really critical. My feeling is that it isn't (you have to start somewhere), and that the contribution is still interesting. This point is highlighted by the authors as a direction for future work - the degradation due to the lower number of observations is never quantified. Non-asymptotic results might be hard to derive, but it is the key argument for not splitting, and I found it a bit frustrating not to have some quantitative insights on this (apart from experiments) - l.140 "The rule for selecting the test statistic from these sets is simply to select the one with the highest value": I think it could be worth discussing/motivating a bit more this choice. In particular, why does the maximum over the norm-1 vectors does not coincide with the maximum on the e_j? - recent works have proposed robust versions of MMD leveraging the Median-of-Means estimator (MONK – Outlier-Robust Mean Embedding Estimation by Median-of-Means, Lerasle et. al. 2019), can the proposed approach be extended to this setting?

Correctness: Yes, I did not carefully check all mathematical proofs

Clarity: Yes

Relation to Prior Work: Yes, the ideas seem novel to me, although I might not have enough knowledge about this precise literature

Reproducibility: Yes

Additional Feedback: Minor comments ******************* - l.55 the observed test statistic \hat{\tau} is never really defined, in particular w.r.t. \tau as r.v. VS realization. This might be a bit confusing - I think Lemma 1 needs rephrasing. In particular, by definition the expectation of \tau is already that of h, so one does not need to assume anything here. And the variance of \tau is \sigma^2 / n. Since \tau is already defined in the previous paragraph, writing "Let \mu denote E[h] and \sigma^2 = Var(h)" seems enough - missing \mid H_A in eq 1? - l. 137 the e_j vectors are not defined, although it remains understandable. Writing T_base = {\tau_1, \ldots, \tau_d} could be more clear - l. 164: allow *us to* - l. 150/204: TN refers to two (slightly) different objects - in Thm 1, the notation =^d is not consistent with the \sim used in Cor 1 Overall evaluation ********************* This work is well presented and the motivation interesting. My only doubt regards the limitation due to the restrictive setting of linear combinations among base test statistics. I am ready to fully support acceptance (7 grade) if this point is clarified either through the rebuttal or discussion with other reviewers Post-rebuttal *************** I have read the rebuttal and other reviews, that partly answered my questions. I am not too concerned with the empirical evaluations, as the theoretical contribution is already consistent. In such cases, experiments showing that the developed methodology "works" (on reasonable datasets) seem enough for me. One interesting point that could be added to a revised version: is there any intuition that could allow to anticipate the truncated behavior of the asymptotic distributions? It seems non trivial, but might help readability (R4). Also, I think the rebuttal could have been more detailed (difficulties pointed by reviewers are only deferred to future work; some basic additional experiments could have been provided), instead of constantly recalling what we have read in the reviews. Nonetheless, I have raised my score to 7 as I think this contribution deserves acceptance.


Review 4

Summary and Contributions: The paper suggests a Kernel based hypothesis testing method where there is no need to split the data into train and test in order to learn the parameters of the model. Specifically, a correction is learned to adjust the threshold and allow for all of the data to be used for training.

Strengths: The paper is interesting and somewhat novel.

Weaknesses: The significance is not apparent since the computational expense is not clearly compared to the case of splitting the data. Using all of the data leads to more power but also more computation during training.

Correctness: I have some doubt about the correctness of the method which might stem from having questions about the derivation but then this is because the paper is not at all clear about the derivation and does not provide any intuition on what is being done in order to achieve the results they claim. I provide a few examples: (1) In line #111 of the paper they mention that P(T>t|A,H)=P(T>t|H) for the splitting method and that is why splitting works and makes statistical sense. This is great, but where does the paper show a similar clear cut statement for the proposed method? (2) In line #136 of the paper it is mentioned that "to select the test statistic we maximize the test power ...." The fundamental problem with this statement is that the thresholds are chosen based in order to get a set \alpha type-I error. The power of the test comes later as a result and should not be used to set the thresholds of the test. (3) The paper uses linear and polynomial kernels in the experiments. These kernels are not universal and hence are not a good choice compared to a universal kernel like the Gaussian.

Clarity: The paper makes a powerful claim that there is no need to split the data into train and test. But the derivation is not at all clear. There is no intuition on why the "correction" should work and what is the main statistical basis for this. It might be the case that I am unfamiliar with certain prerequisites but then these should be explained in the paper. The paper spends a whole page explaining the basic kernel hypothesis testing problem but then fails in its main job of explaining the derivation well.

Relation to Prior Work: yes.

Reproducibility: No

Additional Feedback:

[Author Response · NeurIPS 2020]

We thank all reviewers for their thoughtful reviews. We are grateful that they conclude that "theoretical results of this paper are solid" (R2), the "experiments are conclusive" (R3) and reproducible (R1), and the paper is clear and well written (R1,R2,R3). Below we respond to the reviewer's major concerns and take our responses and the remaining concerns into consideration when revising our paper. Since three reviewers already favor acceptance, we hope that the general skepticism of R4 can be addressed in the discussion phase.

**Applicability beyond linear combinations of base test statistics (R1, R3).** As we discuss in the conclusion of the submission, specifying the base set of test statistics beforehand is indeed a limitation of our approach. Nevertheless, we have two further comments on this. First, this setting is already flexible enough for many practical applications. One can choose the base test statistics in any creative way as long as it does not depend on the data, e.g., a grid of hyper-parameters, domain-specific kernels, etc. Second, a necessity for our approach is the characterization of the *selection event*. For the considered methods, this is doable because we can solve the optimization problem leading to the optimal test statistic either in closed form or via convex optimization. More general approaches, like continuous optimization of a bandwidth or learning a deep kernel, are not convex problems. To the best of our knowledge, the characterization of the selection event under these scenarios remains an open question.

**Motivation and connection to data splitting (R2, R4).** It is important to note that we are not suggesting a new test statistic per se. Linear combinations of linear-time MMD estimates have been considered before, e.g., in reference [4] of the submission. For any (normalized) linear projection $\tau_{\boldsymbol{\beta}} = \boldsymbol{\beta}^\top \boldsymbol{\tau}/(\boldsymbol{\beta}^\top \Sigma \boldsymbol{\beta})^{\frac{1}{2}}$ of the base test statistics $\boldsymbol{\tau}$, where $\boldsymbol{\beta}$ is independent of $\boldsymbol{\tau}$, the asymptotic distribution is standard normal (p.3 in the submission). If the independence of $\boldsymbol{\beta}$ and $\boldsymbol{\tau}$ holds, the asymptotic test power can be derived in closed form for any level $\alpha$, see Eq.(1) in the submission. To guarantee this independence, previous work used data splitting to estimate the optimal combination $\boldsymbol{\beta}^*$ at the expense of the resulting test power. For us, the motivation is exactly the same, except that we aim to avoid data splitting to achieve higher power (see our experiments). Thus our main contribution is to explicitly correct for the dependence between $\boldsymbol{\beta}^*$ and $\boldsymbol{\tau}$ when the same data is used to estimate both of them. In the revised version, we will make this intuition more understandable, and we will also include a proof sketch of Theorem 1 in the main part.
On the other hand, please note that R1 concludes that the "*proofs look sound*" and that our methods correctly control the Type-I error at the rate $\alpha$ (Fig. 7 in the appendix), which is strong empirical evidence that the "*methodology is sound*," as noted by R3.

**Beyond linear-time MMD two-sample tests (R1, R2):** Our methods only require asymptotic normality of a set of base test statistics. This is also the case for the B-test (R1) as well as for goodness-of-fit tests (R2) based on the linear-time kernel Stein discrepancy (KSD). However, asymptotic normality under $H_0$ does not hold for the FSSD given in Jittkritum et al. (2017). Due to the space constraints and to highlight our main contribution, we decided to focus on the simplest case of linear-time MMD estimates and not compare against B-tests or complete U-statistics MMD. We will add a discussion of methods that balance computational efforts with accuracy and the applicability of our methods in the revised version.

**Quantification of the advantage over data splitting (R3):** The central motivation of our work is indeed that data splitting leads to a decrease in test power. Our main contribution is to show that it is *possible* to learn then conduct the test effectively without resorting to data splitting. In the experiments, we show that our approach outperforms data splitting on all splitting ratios, which was perceived as conclusive by R1,R2,R3. The theoretical characterization of this gain will require partial (if not full) knowledge of *finite-sample* (non-asymptotic) distributions of the statistic under both $H_0$ and $H_1$. When the underlying data distribution is unknown (nonparametric setting), these distributions are non-trivial to obtain and remain an open question even in the case of the widely studied T-test statistic. We leave this important direction for future work.

**Others:** - Computational cost (R4): The computational cost of our approach and data splitting are of the same order. The cost of estimating the test statistic and covariance matrix at once with the entire data set is $\mathcal{O}(n)$, whereas data splitting costs $\mathcal{O}(cn) + \mathcal{O}((1-c)n)$ for $c \in (0, 1)$. Thus at the same computational cost, we achieve higher power as shown in our empirical results.

- "*These [used] kernels are not universal*" (R4): This is a misunderstanding. We actually use various Gaussian kernels for all experiments in Fig. 1 which we report in lines 270-272 of our submission.

- "*How [can one] utilize their own data*" (R1): Thanks for this suggestion. We will update the README to enable an easy integration of new dataset. We are thankful that R1 invested the time and managed to reproduce our results.

- l.140 "The rule for selecting the test statistic from these sets is simply to select the one with the highest value" (R3): Note that we defined $\tau_{\boldsymbol{\beta}}$ including a normalization (thus all $\tau_{\boldsymbol{\beta}}$ have unit variance). Hence, the SNR coincides with the value of the test statistic and determines the power. Thus selecting the one with highest value is sufficient. Also note that the norm constraint on $\boldsymbol{\beta}$ is only to ensure a unique solution, see line 121 in the submission.

[Meta-Review · NeurIPS 2020]

Good paper on an important topic. The reviewers found the theoretical work sound. In the final version, the potential extensions of the method to other frameworks could be addressed.